# Solute trapping and non-equilibrium microstructure during rapid solidification of additive manufacturing

Neng Ren [1], Jun Li [1]✉, Ruiyao Zhang[2], Chinnapat Panwisawas [3]✉, Mingxu Xia[1], Hongbiao Dong[4] & Jianguo Li[1]

Solute transport during rapid and repeated thermal cycle in additive manufacturing (AM) leading to non-equilibrium, non-uniform microstructure remains to be studied. Here, a fully-coupled fluid dynamics and microstructure modelling is developed to rationalise the dynamic solute transport process and elemental segregation in AM, and to gain better understanding of non-equilibrium nature of intercellular solute segregation and cellular structures at sub-grain scale during the melting-solidification of the laser powder bed fusion process. It reveals the solute transport induced by melt convection dilutes the partitioned solute at the solidification front and promotes solute trapping, and elucidates the mechanisms of the subsequent microstructural morphology transitions to ultra-fine cells and then to coarse cells. These suggest solute trapping effect could be made used for reducing crack susceptibility by accelerating the solidification process. The rapid solidification characteristics exhibit promising potential of additive manufacturing for hard-to-print superalloys and aid in alloy design for better printability.

Laser powder bed fusion (LPBF) approach offers high-precision fabrication using melting-based additive manufacturing (AM) technologies, and is gradually being adopted by aerospace, automotive or even medical industry to make commercially viable parts thereby disrupting the traditional manufacturing industry[1–4]. Although LPBF shows great potential in industrial applications, the process can still induce defects, such as porosities[5,6], keyholes[7,8], and cracks[9,10], which limit the promotion of innovative technology. The cracks in the printed nickel-based superalloys, in particular, have afflicted the researchers and engineers for decades. Cracks greatly accelerate the rupture of the superalloy components under the high-temperature and high-pressure working condition, and should be inhibited to prolong the creeping rupture life. However, they are always formed even just after printing or heat treatment. Solute segregation directly determines the residual liquid film formed during solidification and the liquation in the

heat-affected zone, which are considered to be root causes of the cracks in the built parts. However, it remains to be studied how the solute elements are transferred during the non-equilibrium rapid and repeated thermal cycles in AM. It is believed that there should be a critical process window for the AM of superalloys, and researchers have conducted parameter study to search for the perfect operating parameters[11–13]. Considering that superalloys exhibit vastly different crack susceptibilities[14,15], others argue that composition adjustment[16,17] and alloy design approach[18–20] should be the better way for the AM of superalloys. Consequently, insight into the solute transport process is demanded to better understand, control, and ultimately eliminate the crack defects.

The cellular-dendritic microscale, extreme high-temperature, and ultra-high-speed process of LPBF brings great challenges to the in-situ experimental observation of the physical phenomenon, especially the

---

[1]Shanghai Key Laboratory of Advanced High-temperature Materials and Precision Forming, School of Material Science and Engineering, Shanghai Jiao Tong University, 200240 Shanghai, P. R. China. [2]Centre of Excellence for Advanced Materials, 523808 Dongguan, China. [3]School of Engineering and Materials Science, Queen Mary University of London, London E1 4NS, UK. [4]School of Engineering, University of Leicester, Leicester LE1 7RH, UK. ✉e-mail: li.jun@sjtu.edu.cn; c.panwisawas@qmul.ac.uk

solute transport and microstructural evolution in the melt pools. In this case, high-fidelity physics-based numerical modelling can be a powerful tool to capture more of the nature of the process. Representative studies[21–23] have provided impressive references for assessing the grain structure of the additively manufactured parts. However, most AM microstructure simulations focus on capturing the grain structure using thermal history only[24,25], and lack of the elemental distribution consideration which is the essence of the etched microstructure. Elemental segregation also influences heterogeneity of material properties and post-process treatments. The transport and distribution of solute elements at the microstructural "sub-grain" scale ($10^{-6}$-$10^{-7}$ m), including the solute partitioning between the cellular (dendritic) trunk and the intercellular (interdendritic) region, as well as the non-equilibrium nature of the solidification process, are not well understood by the previously reported model frameworks[26,27]. Besides, as uncovered by X-ray synchrotron radiation imaging experiments[28,29] and macro-scale simulations[30,31], the melt flow during AM process can make significant impact on the solute transport and hence presumably alters the as-printed microstructure[32–34]. In the recent progress on the modelling of AM[26], the factors including the effect of melt convection and solute partition on the solute distribution, and thus on the crystal growth, have always been neglected in the multi-grid method[35,36] and the microstructural simulation under characteristic thermal conditions[37,38].

In this paper, we develop a fully coupled mathematical model, where the thermal-fluid-solutal-microstructure multiple physical fields are directly described at the sub-grain scale. With the help of the high-fidelity modelling, we reveal the solute transport and the elemental segregation at the cellular scale during the melting-solidification process of LPBF, which are currently difficult to be captured with the available in-situ X-ray imaging and numerical simulations. In particular, systematic analysis of the specific solidification behaviours elucidates and verifies the underlying non-equilibrium nature of microstructural transition, i.e., planar to ultra-fine cellular and then to coarse cellular. Besides, the roles of melt convection on the solute segregation are also demonstrated. Except for the intuitive understanding of the multiple physical fields, the study also provides insight into the printability of different classes of superalloys in terms of the segregation nature, explores the potential novel technical route to reduce microstructure heterogeneity, crack susceptibility, and aid in alloy design for better printability.

## Results

### Thermal-fluid-solutal transport

We use the model framework developed and validated detailed in the Method section, to simulate two-track two-layer LPBF process for three different classes of nickel-based superalloys, which are γ′ precipitation hardening hard-to-print CM247LC, γ″ precipitation hardening easy-to-print Inconel 718, and intermediate γ′ precipitation hardening superalloy ABD-850AM specially designed for AM. The chemical compositions of the superalloys are listed in Table 1.

Figure 1 demonstrates the melt convection, temperature field evolution, solute transport, and sequence of melting and solidification during the LPBF process of ABD-850AM. In addition, Supplementary Movie 1 provides a better view of the process in a more intuitive way. Segregation index (SI), the relative deviation from the initial

composition $SI = (C\text{-}C_0)/C_0$, also known as the normalised solute concentration, is used to quantitatively characterise local solute mass fraction and cellular microstructure. It should be noted that the superalloys are simplified as equivalent binary systems according to the method illustrated in Supplementary Methods. The SI of the equivalent binary system indicates the overall segregation level, which reflects the combined effect of the segregation of all the considered elements.

During the melting process, the recoil pressure greatly depresses the free surface, and drives intense melt flow at the bottom of melt pool (Fig. 1a). After the substrate and the powder are melted, the solute in the original dendritic trunks and the interdendritic regions is well mixed with each other under the strong melt convection (Fig. 1b). As the laser beam moves away, the temperature of the free surface drops significantly. Although the recoil pressure and Marangoni force no longer provide sufficient driving force, the residual momentum of the liquid metal is weakened rather slowly. Thus, the melt flow in the melt pool is still rather intense, as shown in Fig. 1c. The trajectory of the melt convection can be reflected by the solute distribution in the melt pool, also indicating that the rejected solute at the non-planar solidification front is transferred away under the effect of strong residual melt convection, as shown in Fig. 1d. Until the end of the solidification process, the melt convection is still dominated by the residual flow momentum, while the natural thermal-solutal convection makes little influence on the flow pattern in the melt pool (Fig. 1e). The solute transport induced by the melt convection also changes the local composition, leading to "macroscale" negative segregation (SI < 0.0, that is, for the solute element whose partition coefficient is smaller than 1, the local solute concentration is lower than the initial one, and vice versa) in the bottom of the printed layer, as shown in Fig. 1f. Since the free surface of the melt pool is greatly affected by the powder distribution under the effect of surface tension, the melt pool is asymmetrical and hence the last-to-solidify region is offset from the centre of the melt pool to the left border of the printed track.

For the second track of this layer, the protruding part of the previously printed track is melted (Fig. 1g), and the solute element in the second melt pool can also be well mixed during the oscillation of the free surface, but the melt pool melts a larger region than the first track due to the residual heat. Also, it should be noticed that the solute distribution in the bottom of the newly printed track is almost identical to that in liquid metal, as shown in Fig. 1h. The convergence of the cells growing from the border of the track and the ones growing vertically leads to local solute enrichment in overlapping zones, as shown in Fig. 1i.

Figure 1j-o demonstrates the solute transport process during the two-track melting and solidification of the second layer. It was assumed that the first printed layer could be sufficiently cooled to room temperature before the laser power input for the second layer. For both the tracks, most of the first printed layer is melted except for the bottom thin layer (Fig. 1j, m). However, due to the uneven surface of the first layer, the second-track melt pool of the second layer migrates to the right and the protruding part of the first layer is not completely fused, as shown in Fig. 1m, n. Voids or porosities are prone to be formed in the boundary regions of the tracks, due to the insufficient smearing of the melt pool-free surface on the previously printed layer and the unfused powder (Fig. 1k, l).

**Table 1 | Chemical compositions of nickel-based superalloys CM247LC[16], Inconel 718[69], and ABD850-AM[19] (weight %, Ni balanced, minor elements of C, B, and Zr are not listed)**

| Elements | Al | Co | Cr | Fe | Hf | Mo | Ta | Ti | W | Nb |
|---|---|---|---|---|---|---|---|---|---|---|
| CM247LC | 5.71 | 9.24 | 8.62 | 0.02 | 1.37 | 0.54 | 3.08 | 0.73 | 9.93 | – |
| Inconel 718 | 0.43 | – | 19.00 | 18.30 | – | 3.00 | – | 1.00 | – | 5.00 |
| ABD850-AM | 1.29 | 17.60 | 18.68 | – | – | 1.89 | 0.44 | 2.22 | 4.74 | 0.60 |

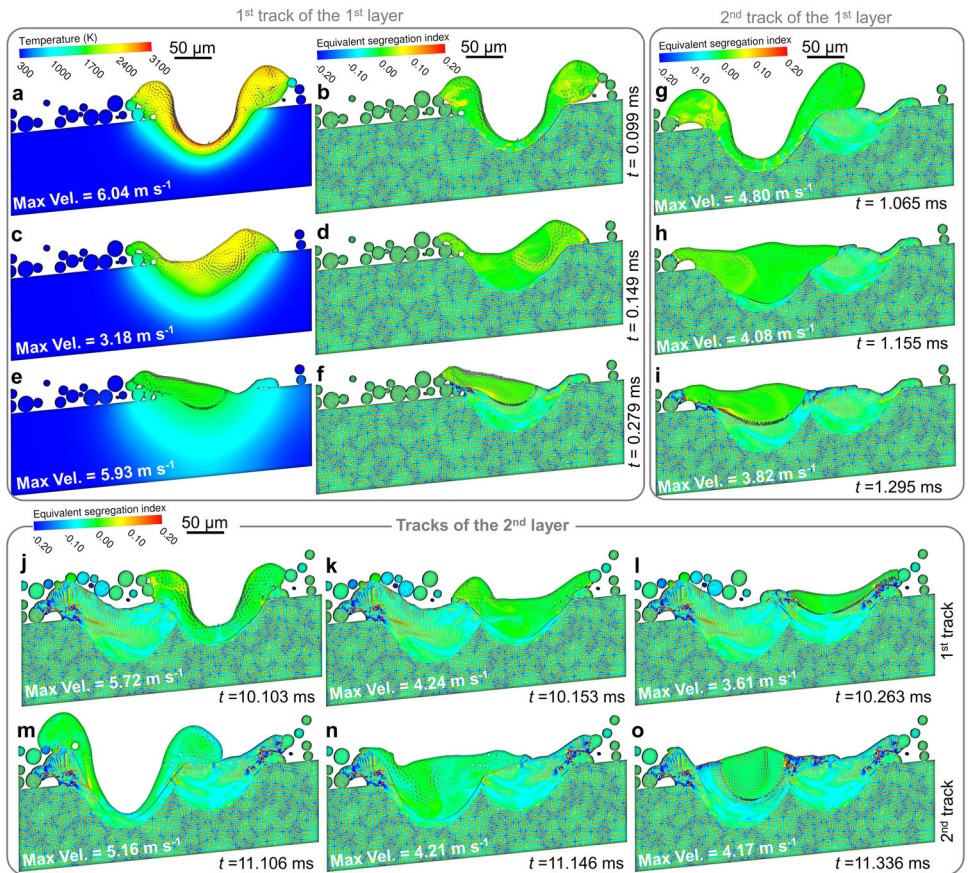

**Fig. 1 | Evolution of thermal-fluid-solutal-microstructural multiple physical fields during the melting and solidification of the two-track two-layer LPBF process of ABD-850AM.** Supplementary Movie 1 provides a more intuitive representation of the transport process of the coupled thermal-fluid-solutal multiple physical fields. **a–f** Simulated thermal-fluid melt pool evolution and the concomitant solute transport during the melting-solidification of the first-track printing of the first layer at 0.099 ms, 0.149 ms and 0.279 ms, respectively. **g–i** Solute transport process and the solid/liquid interface evolution during the second track printing of the first layer. The snap shots at 1.065 ms, 1.155 ms, and 1.295 ms are used to show the melt pool depth, planar interfaces, and cellular structures. **j-o** Evolution of the solute field during the two tracks of the second-layer LPBF. It was assumed that the positions of the laser spots were the same as those of the first layer. Max. Vel. means the maximum flow velocity in the melt pool. See Supplementary Figure 1 for the close-up views of the flow field and solute distribution in the melt pools.

Different from the dynamic moving state of the first layer melt pools, cellular structures develop well from the border and bottom to the centre in the stable melt pools in the final stage of the solidification, as shown in Fig. 1o. In this stage, the melt flow is rather weak and cannot make a significant influence on the solute transport in the liquid metal.

As shown in Fig. 1, the solute transport caused by melt convection brings interesting aspects to the microstructural evolution in AM. Figure 2 demonstrates more characteristics of microstructural transition under the effect of melt convection, and re-melting-solidification induced microstructural re-built in ABD-850AM. During the solidification of the first-track melt pool, the solute enriched in the boundary region of the track can be transferred to the solidification front by the melt flow, resulting in the extra constitutional undercooling at the solid/liquid interface. Consequently, the planar-cellular morphology transition is facilitated (Fig. 2a, b), and cells start to develop sufficiently at the two-third depth of the melt pool. Melt convection can also dilute the enriched solute at the solidification front, and hence leads to a sudden increase in the solidification rate. Therefore, the cellular structure transits back to the planar under this situation (Fig. 2b). The variation of melt flow direction alters these two effects, resulting in the alternating planar-cellular structure. Likewise, the non-uniform solute distribution determines the corresponding local cellular structure morphology. In the solute-enriched region, cellular spacing gets wider, and solute segregation in the intercellular region is more severe, as shown in Fig. 2c.

Due to the solidification sequence and the spreading of the free surface in the powder particles, the solute in the left border of the first track is heavily segregated around the voids between the built regions and the powder (Fig. 2d). However, the remelting of the overlapping zone provides an opportunity for the re-built of the region. It greatly evens the original solute segregation under the sufficient solute mixing effect of melt convection, as shown in Fig. 2e, f. The solidification sequence is completely changed and there is no more powder affecting the surface morphology of the re-built region, as shown in Fig. 2g, h. The severe intercellular solute enrichment (Fig. 2d) in the overlapping zone is significantly improved after the re-solidification (Fig. 2i).

**Solute trapping region and microscale cellular structures**
AM is related to repeated rapid cooling and heating of the new layer on the previous layer with the ultrahigh cooling rate of $10^5 - 10^7$ K/s. The phenomena of re-melting and re-solidification including intrinsic heat treatment (cyclic re-heating the layers) can also occur. So, it is of great interest to examine what happens in the two-track printing of the second layer. Besides, it should be noticed that Fig. 2 shows regions where the solute segregation is so slight that no obvious cellular structures can be observed even at the micrometre scale. According to the fundamental rapid solidification[25,39], we believe (quasi) solute trapping occurs in these regions. It deserves paying special attention to the non-equilibrium behaviours in different types of superalloys, and discussing whether we could make use of this phenomenon to

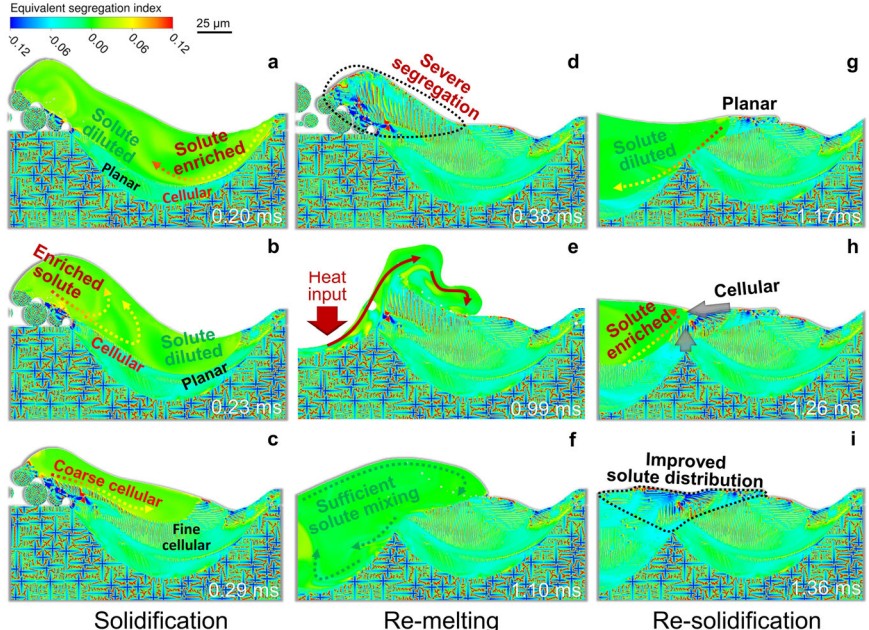

**Fig. 2 | Melt convection induced solute transport and dynamic structural evolution in the melt pool and the overlapping zone in ABD-850AM. a, b** Solute enrichment induced planar-to-cellular structural transition, and solute dilution induced reverse cellular-to-planar transition. **c** Solute distribution and the corresponding local cellular structure. **d–f** Remelting-induced solute transport in the new melt pool. **g–i** Effect of solutal transport on microstructural evolution in the overlapping zone.

reduce crack susceptibility in terms of solute segregation. It is also crucial to confirm whether the non-equilibrium behaviour could be still reserved after the multi-layers printing.

Figure 3 demonstrates the SI distributions of the specific elements in CM247LC (Hf) and Inconel 718 (Nb), and the equivalent solute element in ABD-850AM, since Nb in Inconel 718 and Hf in CM247LC exhibit higher tendency to segregate during the alloy solidification. For the superalloy ABD-850AM, the simulated non-equilibrium nature of microstructural characteristics, including the solute trapping region as well as the subsequent transition to ultra-fine cells and coarse cells, could reach a reasonable agreement with the experimental observations reported by Tang et al.[19], as shown in Fig. 3a–c. During the rapid solidification process, the rapid solid/liquid interfacial velocity induced by the high cooling rate contributes to the solute trapping effect and the planar interfacial morphology[39–41]. Although there are still rather thin striated segregation patterns and fine cellular structures after zooming in on Fig. 3a, c, we could still consider (quasi) solute trapping occurs in this region due to the low local segregation index.

Similar solute trapping regions can also be found in the bottom of melt pools of Inconel 718 (Fig. 3d, e), whereas they are not formed in the melt pools of CM247LC. Instead, in the printed tracks of the superalloy CM247LC, the cellular structure develops sufficiently even in the bottom of the melt pool (Fig. 3g, h). The simulated microstructural characteristics of LPBF-ed Inconel 718 and CM247LC are both in acceptable consistent with the as-printed structures in previous experimental observations shown in Fig. 3f, i. Compared with Inconel 718 and ABD-850AM, the solute SI of CM247LC is much higher, and the rejected solute, especially Hf, is significantly enriched in the intercellular region, particularly where the cells growing along different directions converge. Severe segregation along these boundaries prolongs the solidification process, and creates great resistance for the melt to feed the intercellular solidification shrinkage, thus vastly promoting solidification cracking (Fig. 3f). The accumulated solute elements also lower the solidus temperature of the intercellular regions, leading to the formation of low melting point phases (e.g., carbides and borides) or eutectic phase, hence increasing the risk of liquation

cracks during the printing of subsequent layers. Additionally, the enriched Nb also promotes the formation of Laves phase in Inconel 718[14,19], which negatively affects the mechanical properties of the build parts.

After the second layer powder is melted and then solidifies, due to the powder distribution and the solute distribution of the first layer, the flow field and the solutal transport process can be much different from those in the first layer. Even though melts with different solute concentrations are strongly mixed under the melt convection, the solute distribution in the melt pool is still not completely uniform. As seen from the Supplementary Movie 1 and Fig. 3 (where the arrows indicate the melt flow direction), the solute-diluted melt of higher solidus temperature sways around the centre of the melt pool, causing the local faster solidification rate and facilitating the consequent solute trapping. Besides, the melt flow along the solidification front can also carry the rejected solute away, and thus the solute trapping regions in the centre of the melt pool are generally negatively segregated. More importantly, these behaviours accelerate the solidification process by raising the liquidus temperature. Especially under a relatively low cooling rate (close to the equilibrium state), melt flow can provide extra crystal growth undercooling degree, which promotes the solute trapping effects and enlarge the solute trapping regions during the same thermal history. Moreover, the distribution of the solute trapping regions in the centre appears irregular as the change of the flow pattern is rather complicated.

## Discussion

The non-equilibrium solute transport and trapping can introduce special aspects to the printed layer. Figure 4 demonstrates the effect of solute trapping on the solidification behaviours and crack susceptibility. Solid-state crack could be attributed to the local stress change induced by the precipitation of $\gamma'$ phase[19,42], and the uneven distribution of $\gamma'$ forming elements can also leads to the heterogeneity of precipitation size and mechanical properties[43]. Figure 4a shows the SI distribution of $\gamma'$ forming elements (including Al, Ta, and Ti)[44] in CM247LC in the heat affected zone, which is defined as the region between the isotherm of 1144K[19] and the iso-surface of $f_s = 1$. Heat

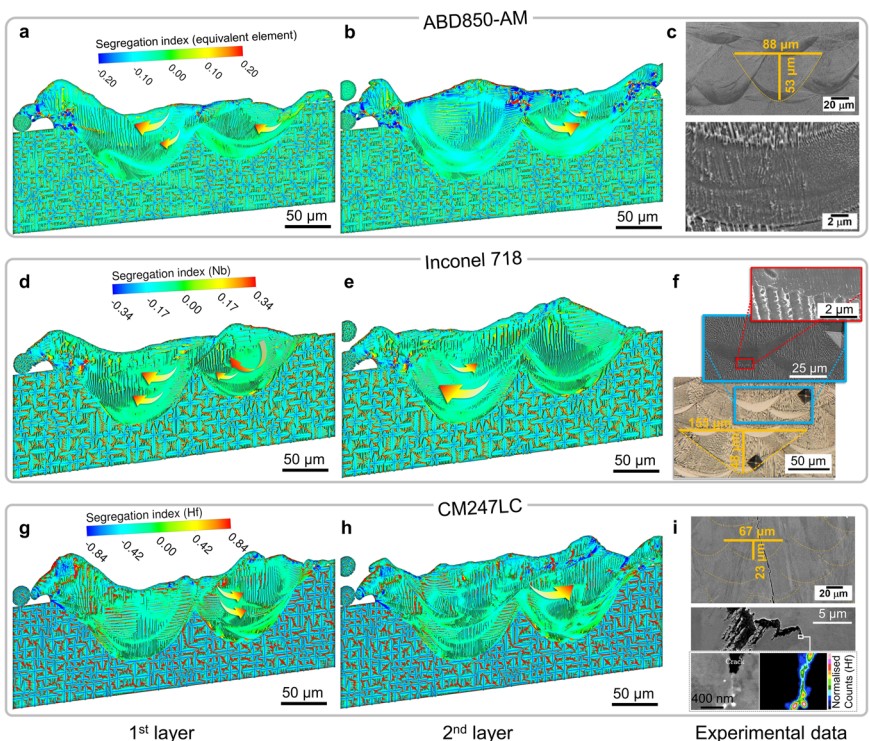

**Fig. 3 | SI distributions in the printed two layers of CM247LC, Inconel 718, and ABD-850AM[19,47,70].** SI of specific elements Nb and Hf are chosen to be plotted for Inconel 718 and CM247LC, respectively. SI distributions are used to characterise the microstructure, since the etched microstructure is actually the result of solute distribution. Iso-surfaces of SI = 0.0 are used to mark the "boundaries" of the cells, and the arrows indicate the flow pattern of the solute-diluted melt. **a, b** The SI distribution of the equivalent solute element of the first (**a**) and the second (**b**) layer of printed ABD-850AM. **c** Dimensions of melt pools and solute trapping regions, and microstructural characteristics of ABD-850AM observed in the experiments[19]. **d, e** SI distribution of Nb of the first (**d**) and the second (**e**) layer of printed Inconel 718. **f** Morphology and size of the solute trapping regions, and microstructural features around the solute trapping region of printed Inconel 718[70]. **g, h** SI distribution of Hf of the first (**g**) and the second (**h**) layer of printed CM247LC. **i** Grain structure, solidification cracks, and measured Hf distribution in the LPBF-ed CM247LC[47]. Different colour maps (SI = −0.2-0.2 for ABD-850AM, SI = −0.34-0.34 for Inconel 718, SI = −0.84-0.84 for CM247LC) are used in the contours for better

view of the cellular structures and segregation patterns. It should also be noted that the experimental results shown in Fig. 3c, f, and i. were observed under different operating conditions (see Supplementary Information). See Supplementary Fig. 4 for the close-up views of the microstructural characteristics observed in these experiments. *Fig. 3i is adapted from Tang, Y. T., Panwisawas, C., Jenkins, B.M., Liu, J., et al. Alloys-by-design: Application to new superalloys for additive manufacturing. Acta Mater. 202, 417–436, Copyright Elsevier (2021). https://doi.org/10.1016/j.actamat.2020.09.023. Figure 3f is adapted from Yi, J. et al. Microstructure and mechanical behaviour of bright crescent areas in Inconel 718 sample fabricated by selective laser melting. Mater. Des. 197, 109259, Copyright Elsevier (2021). https://doi.org/10.1016/j.matdes.2020.109259. Figure 3i is adapted from Ghoussoub, J.N., Tang, Y.T., Dick-Cleland, W.J.B. et al. On the Influence of Alloy Composition on the Additive Manufacturability of Ni-Based Superalloys. Metall. Mater. Trans. A 53, 962-983, Copyright Springer Nature (2022). https://doi.org/10.1007/s11661-021-06568-z. Images were rescaled to improve readability.

affected zones (HAZ) occupies a large region of both tracks in the first printed layer due to the heat transfer caused by the dynamical melt pool, and γ′ forming elements are significantly segregated in the HAZ. It has been revealed that intrinsic heat treatment caused by cyclic re-heating in AM is possible to promote precipitation in HAZ[19,42]. Solute-enrichment of γ′ forming elements is considered to be responsible for the large γ′ precipitates of poor ductility formed in the interdendritic (intercellular) regions[43]. Among the γ′ forming elements in CM247LC, Al is less segregated and the amount of Ti is rather little, so Ta dominates the local composition deviation (the difference from the nominal one, indicated by the grey plane), as shown in Fig. 4b. In contrast, due to solute is much less segregated in the bottom of the melt pool when printing ABD850-AM, the intrinsic heat treatment would make little influence on the potential non-uniformity of precipitated γ′ phase.

Based on the simulation results shown all above, we draw a schematic diagram of the characteristic solute distribution and microstructure of the printed tracks in Fig. 4c. Solute segregation becomes more severe from bottom to top, corresponding to the microstructural evolution from solute trapping region and ultra-fine cells to coarse cells. Higher solute segregation enlarges the difference of solidus temperature between cellular trunks and intercellular

regions, and leads to the more nonuniform mechanical properties due to the differently sized γ′ precipitation phase[43,45]. In this case, the solute trapping behaviours can bring rather different aspects. The low local ductility caused by the large-sized γ′ precipitates can be alleviated by a uniform solute distribution in the solute trapping regions and even in the ultra-fine cell regions. The lower level of segregation caused by solute trapping effect also reduces the grain boundary brittleness to suffer the high thermal stress induced by the thermal cycles[46]. The relation of solute distribution and the cellular structure should be studied to understand the heterogeneity in as-printed parts, and subsequently we can tailor the microstructure using the optimised process conditions and strategic heat treatment protocol.

Compared with the equilibrium solidification process, the freezing range is largely shortened under the non-equilibrium rapid solidification condition (Fig. 4d), which suppresses the solidification crack caused by the inadequate feeding in the final stage of solidification. The raised solidus temperature inhibits the local liquation in the heat affected zone. Furthermore, the great reduction in solute segregation indicates less differences in solidification behaviours and mechanical properties between cells and intercellular regions, so the previously printed tracks carry less risk of being liquated or teared. Reviewing more experimentally observed microstructures and cracks in the

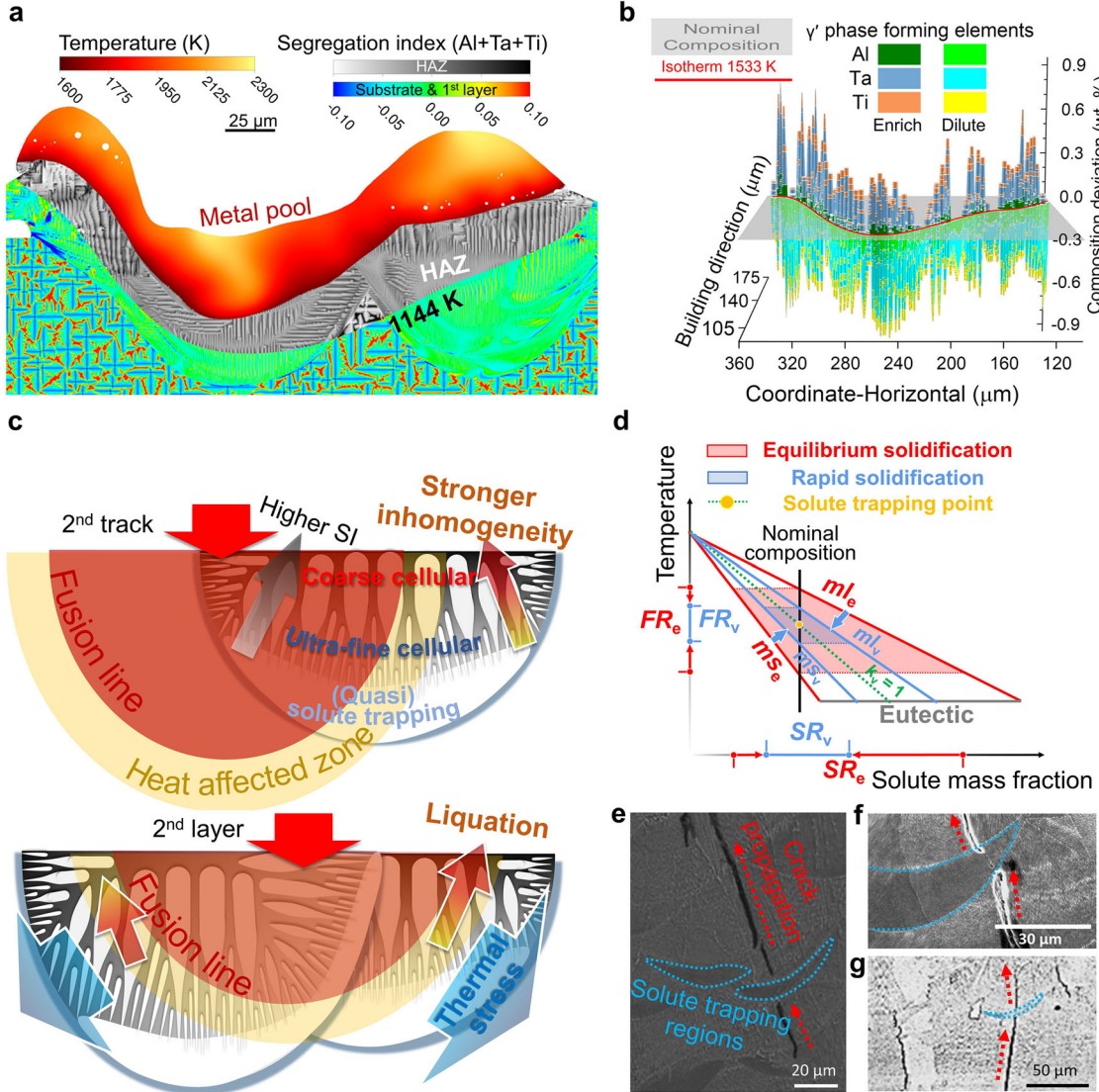

**Fig. 4 | The effect of solute trapping on the solidification behaviours and crack susceptibility. a** Distribution of γ′ forming elements in the heat-affected zone (HAZ). **b** Composition deviation of γ′ forming elements (Al, Ta, and Ti) from the nominal one (the plane of composition deviation = 0) in the HAZ. **c** Schematic diagram of the characteristics of microstructure and segregation pattern. **d** Comparison of the freezing range (*FR*) and segregation range (*SR*) in the phase diagrams under equilibrium and rapid solidification conditions. The outer area depicts the equilibrium solidification process, and the inner one stands for the quasi-solute trapping effect under the rapid solidification condition. The projection on the vertical axis is the freezing range, and the one on the horizontal axis is the segregation range. *ml* and *ms* are slopes of liquidus and solidus, respectively. Subscript e and v indicate the equilibrium state and the rapid solidification condition, respectively. **e**–**g** Suppression of the crack propagation by the solute

trapping observed in the AM of CM247LC[47] (**e**), Hastelloy X[18] (**f**) and Inconel 738LC[48] (**g**). *Fig. 4e is adapted from Ghoussoub, J.N., Tang, Y.T., Dick-Cleland, W.J.B. et al. On the Influence of Alloy Composition on the Additive Manufacturability of Ni-Based Superalloys. Metall. Mater. Trans. A 53, 962-983, Copyright Springer Nature (2022). https://doi.org/10.1007/s11661-021-06568-z. Figure 4f is adapted from Harrison, N. J., Todd, I. & Mumtaz, K. Reduction of micro-cracking in nickel superalloys processed by Selective Laser Melting: A fundamental alloy design approach. Acta Mater. 94, 59–68, Copyright Elsevier (2015). https://doi.org/10.1016/j.actamat.2015.04.035. Figure 4g is adapted from Vilanova, M., Taboada, M.C., Martinez-Amesti, A., Niklas, A., San Sebastian, M., Guraya, T. Influence of Minor Alloying Element Additions on the Crack Susceptibility of a Nickel Based Superalloy Manufactured by LPBF. Materials, 14, 5702, Copyright MDPI (2021). https://doi.org/10.3390/ma14195702.

previous studies[18,47,48], we find it is interesting that though the cracks in the additively manufactured superalloys can continuously propagate to a sufficient length along the building direction, they rarely originate from the solute trapping regions, and even are found to be "interrupted" by the solute trapping region. As shown in Fig. 4e–g, the suppressing effect of solute trapping zone on the propagation of cracks can be found in Hastelloy X[18], Inconel 738LC[48], and even CM247LC[47] which is generally regarded as one of the most hard-to-print or 'unprintable' superalloys. This suggests that the solute trapping effect could significantly reduce the susceptibility of solidification and liquation cracks as well as the potential solid-state cracks, and even

potentially eliminate these crack defects. Furthermore, the ultra-fine cellular structure has also been proved to exhibit better deformation behaviours not only in the nickel-based superalloys (rather stable cyclic response)[49], but also in the high entropy alloys (higher resistance crack propagation and hardness)[50,51] and steels (macroscopically high yield strength)[52].

Figure 5 shows the effect of process condition and common elements in superalloys on the crack susceptibility and qualitative critical process window in LPBF process. Considering that it is generally accepted that the critical window of the process parameters is limited, and that it is difficult to confirm where the window is, researchers

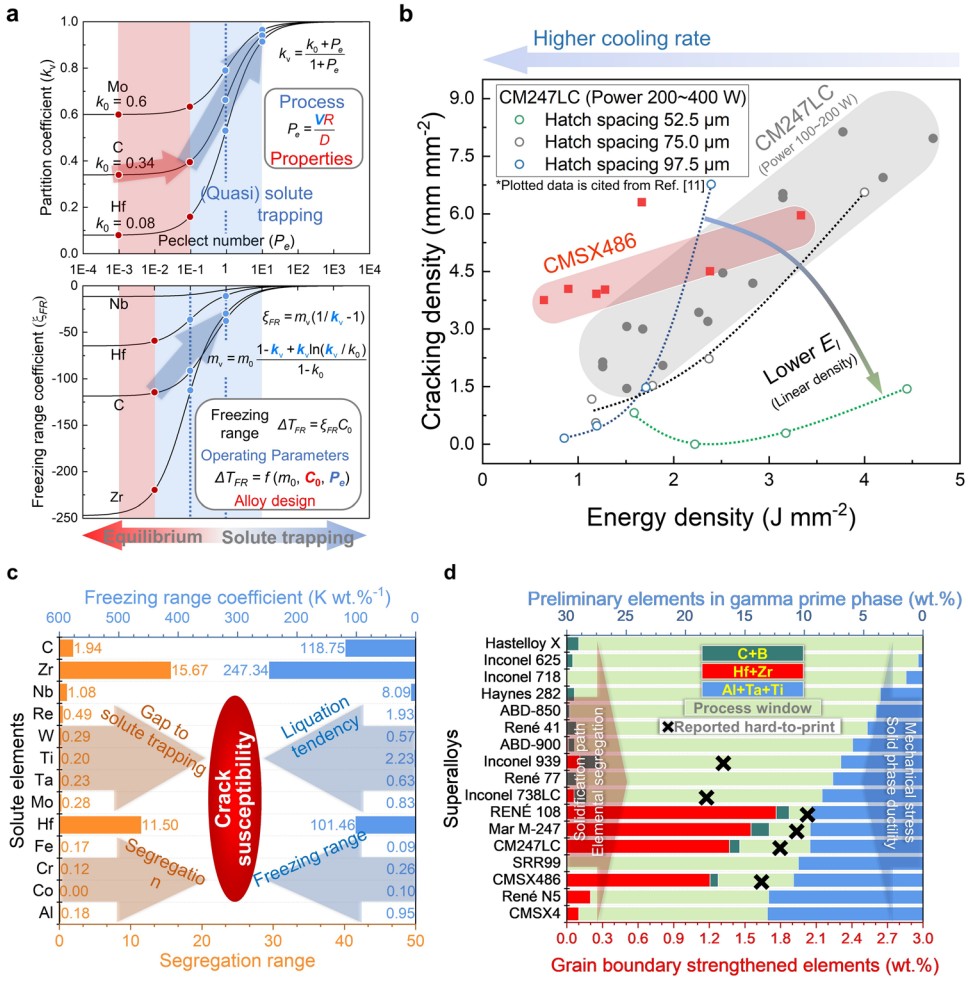

**Fig. 5 | Effect of cooling condition and common elements in the superalloys on crack susceptibility, and qualitative critical process window of LPBF process for the superalloys. a** Effect of crystal growth rate on the solidification behaviours under rapid solidification conditions of AM. Variations of solute partition coefficient and freezing range coefficient with the Peclet number ($Pe$) in the Ni–X binary system (X = Mo, Nb, C, Hf and Zr). $V$ and $R$ are the velocity and width of the solid/liquid interface, respectively. $D$ is solute diffusion coefficient in the liquid, and $m_v$ is the slope of liquid. **b** Maps of the operating conditions and the corresponding cracking densities. The subfigure is redrawn from the data in the reference[11]. $E_l$ is linear energy density. **c** Influence of common elements in superalloys on crack susceptibility. The bottom axis and left bars are used to show the segregation range, and the top axis and right bars show the freezing range coefficients of the solute elements. **d** Qualitive critical process window for the superalloys in AM. The bottom axis is the mass fraction of the grain boundary strengthening elements (bars from the left to the right), including Zr, Hf, B and C, and the top axis is the mass fraction of the main preliminary elements in the γ′ phase (bars from the right to the left), Al, Ta, and Ti, which reflects the volume fraction of γ′ phase. The central bars occupying the remaining space are used to qualitatively estimate the width of the crack-free critical process window in additively manufactured superalloys.

prefer to improve the composition design of the superalloys to make them more applicable to AM[19]. However, adjusting the alloy composition to improve the printability always compromises some of the high-temperature mechanical properties of superalloys. In this situation, promoting solute trapping could be another metallurgical processing route to avoid crack defects. According to the continuous growth model proposed by Aziz et al.[39,41], the effect of local crystal growth rate on the solute partition behaviours is shown in Fig. 5a. Crystal growth Peclet number ($Pe$), which is defined as interfacial velocity ($V$) times interfacial width ($R$) divided by solute diffusivity in the liquid ($D$), is used to describe the relative magnitude between crystal growth and solute diffusion, and $D/R$ is also known as characteristic solute trapping velocity. The increase in solidification rate (interfacial velocity) in the range of $Pe = 0.1$–10 results in a significant improvement in the intercellular segregation, especially for the solute elements whose partition coefficients are far from unity, such as Hf and C, as shown in Fig. 4d. As the result of the local solute partition coefficient approaching unity at the rather high solidification rate, the absolute value of freezing range coefficient ($\xi$) decreases significantly

even in the range of $Pe = 0.01$–0.1, especially for the elements with stronger solute partition effect and greater influence on the liquidus, such as Zr and C.

Figure 5b shows the map of the cracking densities under the corresponding input energy densities[11]. The cracking densities (the length of cracks summed and averaged with respect to micrograph areas) of CMSX486 and CM247LC reduce with the decrease in the input surface or linear energy density. As discussed above, the higher cooling rate obtained by the less energy input promotes the solute trapping effect and reduces the segregation level. This higher homogeneity lowers the risk of the intercellular region liquation in the heat-affected zone as well as the potential solid-state cracking. Moreover, under the higher input of surface energy density, the residual heat accumulated in the built tracks and layers also reduces cooling rates of the melt pools. In some of the superalloys, such as ABD-850AM and Inconel 939[19], and even some hard-to-print superalloys such as CM247LC[47], solute trapping regions can be found at the bottom of the melt pool under certain operating parameters. Therefore, except for the alloy design method, retaining the minor elements for better grain

boundary strengthening effects, using melt pools of smaller volume and lower temperature for high cooling rates could be available approaches for the AM of hard-to-print superalloys. However, simply reducing the input energy density often results in lack of fusion porosity. The better way to increase solidification rate could be the optimisation of heat source profile, such as laser beam design and modulation[53]. The attempts using a doughnut[12] or elliptical energy profile[35], for example, are proved to accelerate the solidification process on the premise of ensuring the fusion.

In terms of alloy design, proposals can also be given based on the different solidification behaviours of the superalloys. It would be easier to reach the solute trapping if the strongly partitioned solute elements were removed. The larger segregation range shown in Fig. 5c means a wider gap to reach solute trapping, indicating that it would be more challenging to reduce and completely eliminate cracks by controlling segregation via optimising the process parameters.

It is still controversial whether precipitation-hardening nickel-based superalloys are definitely unprintable. Though superalloys with high γ' phase volume fraction such as CM247LC and Inconel 738LC are generally considered unprintable, crack-free parts can still be fabricated from superalloys with higher γ' volume fraction, e.g., RENÉ 77[54] and CMSX-4[55]. Also, it should be noticed that the sum of Hf and Zr is high in the reported hard-to-printed superalloys with high volume fraction γ' phase (Fig. 5d), and the cracks are reported to be well avoided in the Hf-free CM247LC[16]. It could be easily deduced that it is actually the elemental segregation that worsens the printing performance of precipitation-hardening superalloys which are sensitive to cyclic thermal stress. As discussed above, from the perspective of solute segregation, we believe that high-volume fraction γ' phase superalloys with less grain boundary strengthening elements could be crack-free after additively manufactured at a high solidification rate. Due to the strong tendency of solute segregation in the cellular trunks and intercellular regions as well as the further impact on solidus temperature, elements such as Hf and Zr are strongly recommended to be removed from the superalloys for AM.

In summary, a two-way fully coupled thermal-fluid-solutal-microstructural mathematical model is developed to understand the dynamic solute transport process and elemental segregation in AM. The results from high-fidelity simulation reveal that the non-equilibrium nature of intercellular solute segregation and cellular structures at the sub-grain scale during the multi-track multi-layer fusion and solidification of LPBF process. The predictions of melt pool, solute trapping zone, and microstructure are well verified by the reported and conducted experiments. Having demonstrated the characteristics of the solute distribution, we elucidate the role of melt convection on elemental segregation and propose a mechanism for the formation of solute trapping region and microstructural evolution in the printed layers. From theoretical perspectives, this work also illustrates a potential technical route of compositional control to optimising operating parameters to reduce crack density or even eliminate cracks in AM of hard-to-print superalloys. Based on the systematic analysis of the specific behaviours of rapid solidification, hard-to-print superalloy can be additively manufactured without removing essential minor elements by further accelerating the solidification to promote the solute trapping, thereby reducing the intercellular solute segregation. By comparing the solidification behaviours of different classes of superalloys, CM247LC, Inconel 718, and ABD-850AM, a qualitative estimate of the crack susceptibility (the width of the critical process window) is developed to aid in the alloy composition design of superalloys. Additionally, the proposed model framework can also be a powerful tool for extended study, such as the in-situ alloying of different elemental powders in LPBF processes for high entropy alloys or high-performance steels.

## Methods

To the best of our knowledge, two numerical methods are generally employed in the simulation of solute distribution and microstructure in AM detailed in the previous publications[26,27]. One is to predict the microstructure evolution and solute distribution using specific temperature gradients and cooling rates, according to the characteristic temperature profile in the melt pool[37,38,56]. Simulated solute distributions and microstructures under different initial and boundary conditions can be used to estimate the regions under corresponding thermal conditions in the melt pool, but the impact of fluid flow is always neglected or only studied using a predefined flow field[21,22]. Besides, the method is only "one-way" coupling, i.e., the temperature distribution and even flow field are exported and assigned as initial conditions to consider their effects on the microstructural evolution, while the solute partition and microstructure are unable to affect the solute redistribution and melt flow in the bulk region. Only one direction of the interactions during the solidification process is described. "Two-way coupling", which means the transport phenomenon (including temperature and solute fields) can determine the microstructural evolution, and the solute partition at the microstructural scale can in reverse change the solute transport in the melt pool, is strongly recommended for more tracks and layers printing (cycled melting and solidification).

The other one is multi-grid method, where child grids are used for the grain evolution at microstructural scale ($10^{-7}$–$10^{-6}$ m) and parent grids for the thermal-fluid-chemical interactions at macro scale ($10^{-6}$–$10^{-5}$ m). It is usually employed to resolve the multiscale phenomenon by interpolating macroscale temperature field to the microscale computational grids, and shows good accuracy and efficiency in predicting the grain structure of the entire component[57–59]. However, in this method, the grain structure is only determined by the temperature field, and the effect of melt convection is actually the effect of melt convection on temperature field. Another rather important factors, the partition and the transport of solute elements have not been directly included in the model framework[34]. Despite the rapid development of models for predicting the microstructure of additively manufactured parts in the past decade, there is still a lack of description of the melt flow-induced solute transport and the solute partition at the microstructural scale. Since the solute transport occurs not only at the macroscale heat transfer[22,23], but also at the microstructural scale, it is difficult for the previously developed models to demonstrate concomitant solute transport during the grain growth.

Here, we develop a "two-way" fully coupled cellular automaton-finite volume method to simulate the heat transfer, fluid flow, solute transport, and microstructural evolution in the LPBF process. In the model framework, cellular automaton is employed to describe the preference growth of cells and dendrites, and finite volume method is used to solve the governing equations of the thermal-fluid-solutal interactions in the multiple physical field transport process. Temperature, flow velocity, solute concentration, and microstructure are directly coupled with each other on the same set of computational grids, which means the cells in cellular automaton are also the control volumes in finite volume method. This also enables the solute transfers between the interdendritic (intercellular) regions and the bulk melt pool. Thus, the "two-way coupling" between dendritic structure and thermal-fluid-chemical multiple fields at the melt pool scale can be realised based on the proposed model framework.

### Fluid flow

Metallic phase and gaseous phase are included in the multiphase framework, and the volume of fluid (VOF) approach is used to track the metal/gas interfaces. Molten metal and solidified metal are distinguished by another variable, solid volume fraction. Consequently, volume averaged physical properties are employed

in the control volumes:

$$\frac{\partial \alpha_m}{\partial t} + \nabla \cdot (\alpha_m \boldsymbol{u}) = 0 \tag{1}$$

$$\phi = \alpha_m \phi_m + \alpha_{gas} \phi_{gas} \tag{2}$$

where $\rho$ is density, $\alpha$ is the volume fraction of a fluid phase, $t$ is time, vector **u** is flow velocity, $\phi$ represents one of the physical properties in each control volume, and subscripts m and gas stand for metal and gas, respectively. With the interface between liquid melt and protective gas tracked, the volume of gas trapped in the liquid melt and unable to escape then becomes "process-induced porosity/void" or "fluid flow-induced porosity/void". Note that the formation of vapour and the subsequent formed gas bubbles are not included in the model framework.

Navier-Stokes equation is used to describe fluid flow, including the smearing of the melted powder and the melt convection. For the metallic phase, the stationary solidified metal is distinguished from the liquid metal by solid volume fraction, and the momentum damping in the solidified metal **F_damp** is calculated using Kozeny–Carmen equation. The density changes induced by temperature and solute concentration are regarded as buoyancy using Bousinesq approximation. The surface tension of the gas/metal interface is loaded as a source term, and a temperature-dependent surface tension coefficient is used to describe the Marangoni effect[60]. Recoil vapour pressure **P_recoil** is also exerted on the gas/metal interface as a body force, but the vaporised metal phase and the mass transfer from the liquid metal to metal vapour are not considered[61].

$$\frac{\partial \rho \boldsymbol{u}}{\partial t} + \nabla \cdot (\rho \boldsymbol{uu}) = \nabla p + \sigma \kappa \nabla \alpha + (1 - f_s) \mathbf{F_b} + \mathbf{F_{damp}} + \mathbf{P_{recoil}} \tag{3}$$

$$\sigma(T) = \sigma_0 + \left(\frac{\partial \sigma}{\partial T}\right) T \tag{4}$$

$$\mathbf{F_b} = \rho_m \alpha_m \boldsymbol{g} \left[ \beta_T (T - T_{ref}) + \beta_C (C - C_0) \right] \tag{5}$$

$$\mathbf{F_{damp}} = \rho_m \alpha_m \mu \frac{5 S^2 f_s^2}{(1 - f_s)^3} \boldsymbol{u} \tag{6}$$

$$\mathbf{P_{recoil}} = 0.54 P_0 \exp\left[\frac{L_v M (T - T_v)}{R T T_{!v}}\right] \nabla \alpha_m \tag{7}$$

where $p$ is pressure, $\mu$ is dynamic viscosity, $\sigma$ is surface tension coefficient, $\kappa$ is the curvature of gas/metal interface, $f_s$ is solid volume fraction, $\mathbf{F_b}$ is thermal-solutal buoyancy, $T$ is temperature, **g** is gravitational acceleration, $\beta_T$ is thermal expansion coefficient, $\beta_C$ is solutal expansion coefficient, $T_{ref}$ is reference temperature, $C$ is solute concentration, $C_0$ is initial solute concentration, $S$ is the solid-liquid interfacial area per volume, $P_0$ is ambient pressure, $L_v$ is the latent heat of vaporisation, $M$ is the molecular mass of metal vapour, $T_v$ is boiling temperature, and $R$ is the universal gas constant,

## Heat transfer
Enthalpy equation is employed to describe the heat transfer during the LPBF process. Laser heat source $q_{laser}$ is assumed to be a Gaussian distribution on the gas/metal surfaces of the powder and the substrate[62]. Due to the complex reflection among the particles, the laser heat source is assumed to be uniformly distributed along the building direction. The effect of the vaporisation of liquid metal on the

temperature field is described with a source term $q_{vap}$ applied on the gas/metal interface[61,63]. The absorbed or released latent heat during the liquid/solid transition $q_{latent}$ is also loaded as a source term in the energy equation. Additionally, the radiative heat loss from the free surface of melt pool $q_{rad}$ is included as a source term in the governing equation as well[36].

$$\frac{\partial \rho \boldsymbol{u} h}{\partial t} + \nabla \cdot (\rho \boldsymbol{u} h) = \nabla \cdot (\lambda \nabla T) + q_{laser} + q_{vap} + q_{latent} + q_{rad} \tag{8}$$

$$q_{laser} = \frac{f_{laser} \xi P}{\pi r_0^2} \exp\left[-f \frac{(z + vt - z_0)^2 + (r - r_0)^2}{r_0^2}\right] |\nabla \alpha_m| \frac{2\overline{\rho c_p}}{\rho_m c_{p_m} + \rho_g c_{p_g}} \tag{9}$$

$$q_{vap} = -0.82 \frac{L_v M}{\sqrt{2\pi MRT}} P_0 \exp\left[\frac{L_v M(T - T_v)}{RT T_v}\right] |\nabla \alpha_m| \frac{2\overline{\rho c_p}}{\rho_m c_{p_m} + \rho_g c_{p_g}} \tag{10}$$

$$q_{latent} = \rho_m \alpha_m L \frac{df_s}{dt} \tag{11}$$

$$q_{rad} = -\sigma_s \varepsilon (T^4 - T_\infty^4) |\nabla \alpha_m| \frac{2\overline{\rho c_p}}{\rho_m c_{p_m} + \rho_g c_{p_g}} \tag{12}$$

where $h$ is enthalpy, $\lambda$ is thermal conductivity, $f_{laser}$ is laser power distribution index, $\xi$ is absorption coefficient, $P$ is laser power intensity, $c_p$ is specific heat, $L$ is the latent heat of fusion, $r_0$ is the radius of laser spot, $v$ is laser scanning speed, $(z, r)$ describes the relative position of the centre of the laser spot, $(z_0, r_0)$ indicates the initial position of the laser spot, $\sigma_s$ is Stefan-Boltzmann constant, and $\varepsilon$ is the emissivity of metal.

## Mass transfer and microstructure evolution
Cellular automaton model is employed to describe the preference growth of crystals including cells and dendrites[64,65]. Grain growth is described by the increase in the solid volume fraction, and vice versa. In the cellular automaton model, three states of liquid, interface and solid are defined. The phase transformation between solid and liquid occurs only in the control volumes of the interface state. Interfacial equilibrium conditions are used to calculate the solid volume fraction change in each timestep:

$$\frac{\partial \alpha_m f_s}{\partial t} = \alpha_m \frac{\Delta f_s}{\Delta t} = \alpha_m \frac{f_s - f_{s0}}{\Delta t} = \begin{cases} \alpha_m f_I \frac{C_I^* - C_I}{C_I^*(1-k)} \frac{1}{\Delta t} & \text{solidification} \\ \alpha_m f_I \frac{C_I^* - C_I}{C_I^* - C_s} \frac{1}{\Delta t} & \text{melting} \end{cases} \tag{13}$$

where the subscript 0 indicates the value of a variable at the previous timestep, the superscript * stands for the interfacial equilibrium condition, and $k$ is solute partition coefficient. The state of the control volume changes to "solid" when the solid volume fraction reaches unity, and becomes "liquid" as the solid volume fraction decreases to zero.

A Gaussian distribution is employed to describe the nucleation in the melt pool[66]. Based on the mean nucleation undercooling and the corresponding standard deviation of the alloy, the critical undercooling degree of each nucleus is randomly assigned. As the local undercooling reaches the critical undercooling, the state of the control volume then changes from "liquid" to "interface". For the state transition during the crystal growth, the state changes from liquid to interface when the dendrite arm is long enough to capture the neighbour volume(s), or the state of one of its neighbours changes to

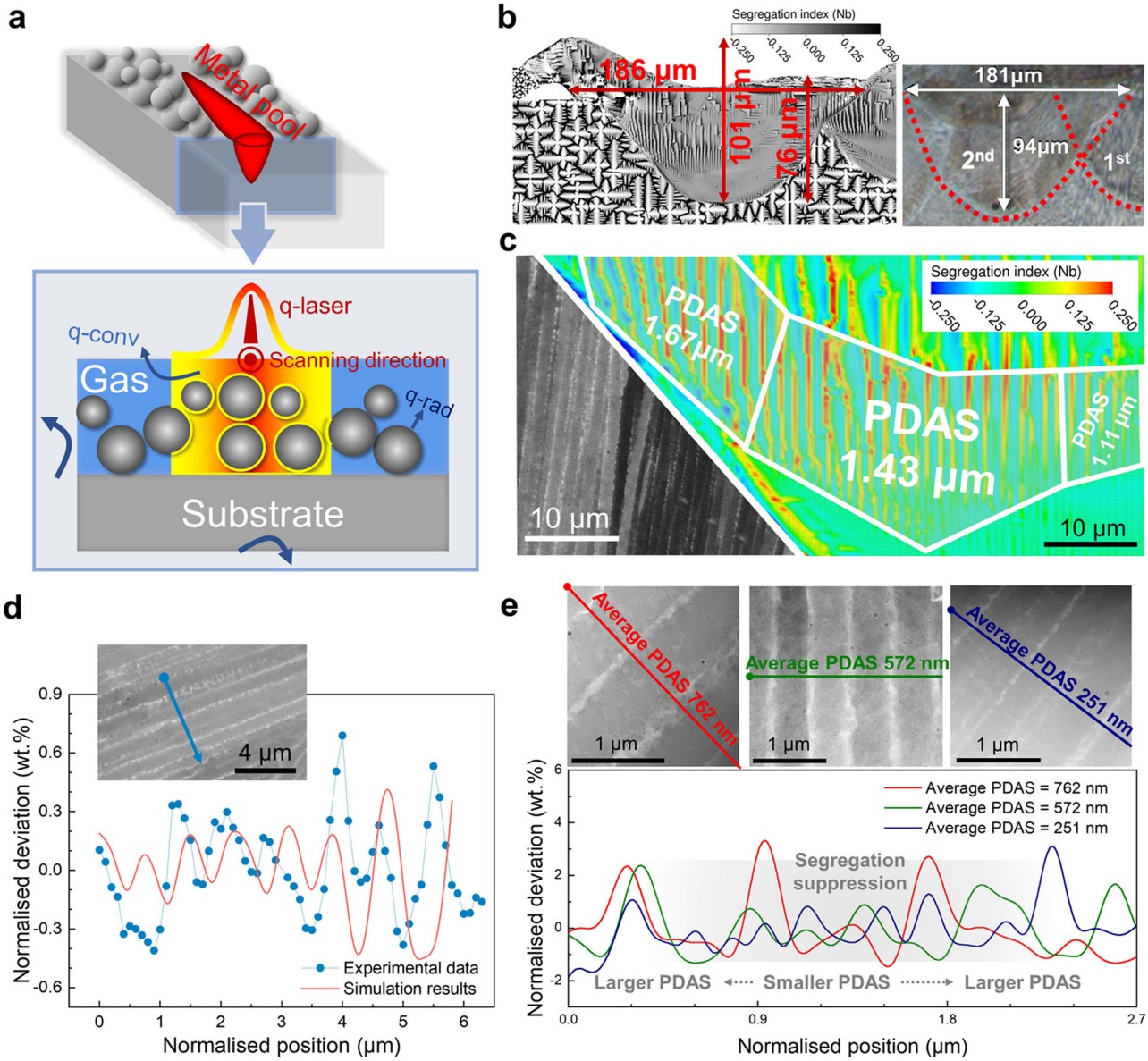

**Fig. 6 | Schematic diagram of computational domain and deterministic conditions, and verification of the simulation results of the benchmark case (Inconel 718). a** Schematic diagram of computational domain, initial and boundary conditions. The two-dimensional computational domain is a longitudinal section of the three-dimensional geometry, including the powder and the substrate. The notes q-laser, q-conv, and q-rad indicate the heat flux on the gas/metal interfaces induced by power input, convective heat loss, and radiation, respectively. **b** Comparison of the size (depth and width) of the second-track melt pool of LPBF-ed Inconel 718 between simulation results and reported experimental data[68]. Since the surface of the printed layer is rather uneven, the quantitative validation is performed using the characteristic melt pool width, depth, and height.

**c** Comparison of experimentally observed and simulated cellular structures, and the distribution of primary dendritic (cellular) arm spacing (PDAS). Solute segregation distribution is used to characterise the microstructure of the printed tracks. Computational grids with the size of 0.25 μm is plotted. **d** Comparison of experimental data (EPMA-WDS) and simulation results on Nb distribution. **e** Effect of solute trapping effect on the segregation profile and primary dendritic (cellular) arm spacing. *Fig. 6b is adapted from Lee, Y. S. & Zhang, W. Modelling of heat transfer, fluid flow and solidification microstructure of nickel-base superalloy fabricated by laser powder bed fusion. Addit. Manuf. 12, 178–188, Copyright Elsevier (2016). https://doi.org/10.1016/j.addma.2016.05.003.

solid. For the melting process, the transition is from solid to interface when the state of one of its neighbours changes from interface to liquid[67].

The solute distribution and the consequent constitutional undercooling determine the growth speed and morphologies of solid/liquid interface, i.e., planar, cellular, and dendritic. Conversely, the solute element rejected from the solidification front during solute partition becomes extra solute source and transfers in the melt pool. Considering the effects above, the solute transport equations in the liquid metal and solidified metal are given by:

$$\frac{\partial \alpha_m \rho_m C_l}{\partial t} + \nabla \cdot (\alpha_m \rho_m \boldsymbol{u} C_l) = \nabla \cdot (D_l \alpha_m \rho_m \nabla C_l) + \begin{cases} \alpha_m \rho_m C_l^*(1-k_v)/f_l \frac{\partial f_s}{\partial t} & \text{solidification} \\ \alpha_m \rho_m (C_l^* - C_s)/f_l \frac{\partial f_s}{\partial t} & \text{melting} \end{cases}$$

(14)

$$\frac{\partial \alpha_m \rho_m f_s C_s}{\partial t} = \nabla \cdot (D_s \alpha_m \rho_m f_s \nabla C_s) + \begin{cases} \alpha_m \rho_m k C_l^* \frac{\partial f_s}{\partial t} & \text{solidification} \\ \alpha_m \rho_m C_s \frac{\partial f_s}{\partial t} & \text{melting} \end{cases}$$

(15)

**Table 2 | Process conditions used in the laser powder bed fusion of the Inconel 718 sample**

| Process conditions | Value |
|---|---|
| Laser power (W) | 285 |
| Scanning speed (m·s⁻¹) | 0.96 |
| Laser spot diameter (μm) | 100 |
| Height of fused layer (μm) | 40 |
| Hatch spacing (μm) | 116 |
| Convective heat transfer coefficient (W·m⁻²) | 80 |
| Emissivity of metal | 0.8 |

where $D$ is solute diffusion coefficient, subscripts l and s stand for liquid and solid, respectively. The melting process can be regarded as the inverse of the solidification process

## Simulation parameters

Figure 6a is a schematic diagram of the computational domain used in the benchmark case. Under the consideration of the computational cost, simulations were conducted in a two-dimensional 210 × 400 μm computational domain, which was the longitudinal section perpendicular to the scanning path. The domain was uniformly divided into 1,341,561 quadrilateral grids with a size of 0.25 μm. Consequently, according to the relative position of the centre of the laser spot and the section, the heat source applied on the gas/metal interfaces in the computational domain was also the projection of the actual three-dimensional gaussian distribution profile on the section.

Initially, we marked the areas of the substrate and the powder, and then set them to be filled with metallic phase with initial composition and liquidus temperature. Next, we simulated the grain nucleation and growth in the substrate and the powder using conventional casting conditions. After the solidification process, the temperature of the domain was reset to room temperature. Thus, we got the solute distribution and grains in the substrate and powder as the initial conditions for the LPBF process. A constant convective heat transfer coefficient and a constant freestream temperature of 300 K were applied at the boundaries of the computational domain. The time interval between the two tracks was calculated according to the scanning speed and the length of the substrate.

## Experimental

In order to further validate the model framework in more aspects, we have also conducted experiments characterising the microstructure and measuring the elemental profile across the cellular trunks and intercellular areas. An Inconel 718 sample of 10 mm × 5 mm × 5 mm (length-width-height) was LPBF-ed on a flat Inconel 718 substate under argon shielding with the process parameters listed in Table 2. The sample was then sectioned longitudinally perpendicular to the scanning direction for characterisation. The microstructure on the section was characterised using In-Beam backscattered electron (BSE) on a field emission gun scanning electron microscope (FEG-SEM) of TESCAN MIRA 3. Quantitative line scans for the elemental profile of Nb were performed on the wavelength dispersive spectrometer equipped on an electron probe microanalyzer (EPMA-8050G, Shimadzu). The sample was then sliced, machined, and grinded into metallic foils with diameter of 3 mm and thickness of 80 μm. After twin-jet electropolishing, microstructural characterisation and the corresponding chemical composition analysis were conducted using scanning transmission electron microscopy and energy dispersive X-ray spectroscopy (STEM-EDS) and high-angle annular dark-field (HAADF) detector on a Talos F200X.

## Model verification: Precision and high-fidelity

Figure 6b shows the simulated solute distribution in the printed layer after the two-track LPBF. The characteristics of the solute distribution in the melted region are quite different from those in the unfused substrate, indicating the profiles of the melt pools. The simulated melt pool profiles are in good agreement with the reported experimental results in ref. 68. Considering the complicated geometry of the newly printed track, the dimensions of the simulated melt pool, including the characteristic depth, width and height, can be considered to be in quantitative agreement with the experimental data[68]. Figure 6c shows the simulated characteristic morphology of the cellular structure is similar to the experimentally observed one. The cellular spacing ranges from 1.25 μm to 1.75 μm, which is comparable to the experimental data from 1.0 μm to 1.8 μm[68].

A wavelength dispersive spectrometer (WDS) equipped on an electron probe microanalyser (EPMA) was employed to quantify the solute profile across the cellular trunks and intercellular areas with the spatial resolution of 100 nm. Figure 6d shows the comparison of the mass fraction profile of Nb, the element of the most focus with strong segregation tendency in the superalloy Inconel 718. It could be concluded that for similar primary dendrite (cellular) arm spacings (PDAS), the simulation results and experimental data reach a reasonable agreement. To further verify the simulated solute trapping effect, the elemental profiles of the cellular structures of different PDASs were characterised using scanning transmission electron microscopy and energy dispersive X-ray spectroscopy (STEM-EDS). Figure 6e demonstrates the Nb concentration distributions across the cellular trunks and intercellular areas. The segregation range, that is the oscillation of solute profile, decreases with the refinement of the cells, which corresponds to the high local solidification rate. Thus, these further verify the simulation results and discussion in terms of the solute trapping effect.

From the acceptable deviations between the simulation results and experimental data, it can be concluded that the proposed model with predictive capability is able to provide reliable predictions for the intended study and further discussion.

## Reporting summary

Further information on research design is available in the Nature Portfolio Reporting Summary linked to this article.

## Data availability

The data that support the findings of this study are available from the authors on request. Sources of the adapted figures from the cited references are provided as a Source Data file. Source data are provided with this paper.

## Code availability

The code that supports the findings of this study is available from the authors on request.

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

## Acknowledgements

This work was sponsored by the National Natural Science Foundation of China No. 52074182 (J.L.), Natural Science Foundation of Shanghai No. 22ZR1430700 (J.L.), and Startup Fund for Young Faculty at SJTU (N.R.). C.P. would like to acknowledge the support from UKRI Innovation Fellowship funded by EPSRC (EP/S000828/2). N.R. would like to thank Prof. Moataz Attallah from the University of Birmingham for providing the raw data of their work. N.R. also thanks Lin Yang, Zhizhuo Wang, Chenyang Hou, Zhiping Wang, Hui Wang, Luwei Yang, and Wanting Tang for their help and discussion. J.L. would like to thank the help from Prof. Changjiang Song. R.Z. would like to acknowledge Changzhou Gangyan Aurora Additive Manufacturing Co., Ltd for the technical assistance on the conducted experiments.

## Author contributions

J.L. and C.P. proposed the original problem and managed the research activity execution. N.R. developed the concept and the methodology. N.R. and C.P. analysed the results and drafted the manuscript with help from all the other authors. R.Z. conducted the experimental investigations. J.G.L. made contributions to the theoretical analysis on the rapid solidification behaviours. M.X. and H.D. reviewed, discussed, and edited the manuscript. M.X. and J.G.L. acquired the financial support for the project. The project was coordinated by J.L., H.D. and J.G.L.

## Competing interests

All the authors declare that they have no known competing financial/non-financial interests or personal relationships that could have appeared to influence the work reported in this paper.
