## [Peer Review File · Nature Communications]

Solute trapping and non-equilibrium microstructure during rapid solidification of additive manufacturingREVIEWER COMMENTS

Reviewer #1 (Remarks to the Author):

The authors have established an interesting numerical solution methodology in a nicely forward looking manner and address with the capability a highly relevant material sciences and engineering problem. Comments and questions to address.

As there are phase to computational fluid dynamics solvers with thermochemical effects, what is the actual novelty of the work in terms of the respective developments?

The term "macroscale grain structure" is used primarily to reflect effects without segregation, please utilize something less ambiguous (p. 4 and 23, for example).

Have you quantified effects with and without convective contributions? A comparison or even some remarks would greatly add to the value and demonstrating the significance of this work.

The non-equilibrium nature is not particularly unexpected here but widely reported in earlier works (line 170 forwards) - in your treatment of, e.g., trapping is there anything you would see differentiating your work from earlier ones?

The binary (dilute-like) treatment of the multi-elemental system, have you made any sensitivity analyses or addressed and investigated any possible effects which might not be captured due to this approximation?

The powder bed looks like it has a rather low fraction of powder, how was this determined and you see this as realistic and not influencing the results?

How was the initial microstructure and simulation initialized?

The analysis is 2D essentially? What about possible implication of 3D effects in the results.

"The microstructure in the overlapping zone is greatly improved after the re-solidification" (line 161)- what is actually improved and where does this relate to?

Solute trapping driven suppression of cracks, can you make any mechanical arguments based on, e.g., the specific behavior of the studied alloys as to why this could be a benefit?

The level of validation in the work, is there validation beyond the given literature sources? Also, the validation seems to point primarily towards bead sizes and shapes, while the results relate to segregation and trapping. Could you provide information on how the method compares, e.g., in terms of the predicted elementary distribution or microstructural features directly?

Reviewer #2 (Remarks to the Author):

The concept of this work is novel and interesting. The idea of coupling melt pool modeling, fluid flow, and solute trapping during additive manufacturing is unique. The work has the potential to enable guidelines for determining printability of materials in a predictive manner rather than the empirical approaches in use currently. However, the paper, in its current forms has several shortcomings that

should be addressed before being considered for publication:

1. The abstract should be modified to highlight the uniqueness of the work as well as key results. In the current form, the abstract appears to be a summary of "what was done" rather than "key findings of the paper". This change will be important to intrigue and excite a broader audience to read the paper
2. The manuscript, in general, is very difficult to read and appears to jump from one concept to another. Certain sentences appear incomplete or unclear. There is use of jargon or overly complex sentence construction. As an example the authors use metal pool instead of the more common term "melt pool". I would suggest the authors to make the manuscript more streamlined and succinct in nature. It will enable the communication of a clear-concise message. In the current form the novel needs to be searched for vs. streamlining it will make it clearer. The attached document has more specific comments and some suggested corrections. As an example the authors use thermal-solutal-chemical-multiple physics as the first section of results. This is quite confusing and does not quite inform the reader, even those familiar with metal AM and modeling, as to what this section is about.
3. In the introduction, the authors claim that the use of convention is still always neglected. This is incorrect since T. Debroy and Wei Zhang have focused on these aspects in their works to name a few. The authors also say that they have developed a two-way fully-coupled mathematical model. Can they clarify, what does two-way mean here?
4. In the results section the authors start with "with the well validated model", yet there is no direct reference to either the supplementary section or a figure therein that can guide a reader to figure out how was the model validated? Where was the validation data obtained from? Granted some of these aspects are discussed later, but this style of writing breaks the flow and raises more questions than it addresses answers.
5. In figure 1, the color map is difficult to read. Further, as the melt pool movement is shown it would be beneficial to also highlight the velocity vectors since it has a significant role in the melt pool dynamics
6. Negative segregation is used as a key concept. However, it is not clear as to what it means or how is it defined? Also, the use of slight negative segregation leaves ambiguity about how much is slight and how much is more. It is recommended that such qualitative assessment be avoided for clarity.
7. In Figure 3, the segregation index data is not normalized and the minimum and maximum values in the color band are not the same so it is difficult to quantify the differences visually. Also, it is recommended to add the reference for the experimental data in the figure caption. While the authors claim that the model agrees with the data, has the model been normalized with respect to the process variables (power, velocity, hatch) used for the experimental data? If not, what is the impact of these variables? Further, it is not clear how the experimental data is corroborating the findings in the absence of discussion on the role of process variables.
8. At the end of the results section, the authors claim that the solute segregation is occurring near the porosity. While one can reach that conclusion from Figure 3i, that is a special case (also, the figure is very hard to read). Most importantly, the authors have not shown or discussed how they have accounted for that in their model? The images from the model seem to show a melt pool with segregation but none of the images have a representation of porosity, lack of fusion or gas porosity. Therefore, at best it is an assertion.
9. In the methods section, while describing fluid flow, the authors call it two fluid phases of metal and gas. It should be simplified to molten metal and gas.

Therefore, in summary, the concept proposed by the authors is unique and significant but there need to be substantial improvements before the manuscript can be considered for publication.

Dear Reviewers:

This is response to reviewers' comments to attach to the resubmission of the revised manuscript entitled "**Solute trapping and non-equilibrium microstructure during rapid solidification of additive manufacturing**". Thank you very much indeed for your thought-provoking comments and suggestions. The specific comments following your suggestions have now been in turn addressed here point-by-point. The revised manuscript is also enclosed with this submission, and it has been substantially improved to take into account the reviewers' comments carefully. The amendments and corrections corresponding to the comments are highlighted in yellow in the revised manuscript. We have shown the changes and reply to your comments (marked with underline) in details in the detailed "response to reviewers' comments", which are as follows:

Reviewer #1:

The authors have established an interesting numerical solution methodology in a nicely forward looking manner and address with the capability a highly relevant material sciences and engineering problem.

Response:

The authors would like to thank the reviewer very much for the compliments and appreciate the valuable comments to improve the manuscript further. We have done our best to address the questions and comments point-by-point carefully.

Comments and questions to address.

1) As there are phase to computational fluid dynamics solvers with thermochemical effects, what is the actual novelty of the work in terms of the respective developments?

Response:

Thanks for your valuable comments and sorry we did not articulate the novelty of the developments on our CFD solvers clearly. To the best of our knowledge and the most recent developments^{1,2} on the high-fidelity modelling of additive manufacturing, in terms of the

thermal-fluid-solutal multiple field transports in a CFD solver, we believe the novelty lies in the development of the strongly coupled model between the thermal-fluid-solutal dynamics and solidification behaviours to gain insight into the thermal-solute interaction, non-equilibrium microstructure, to rationalise the crack susceptibility in next-generation nickel-based superalloys. Multi-grid approach^{3,4} is generally employed in the previous solvers to simulate the grain growth in alloys. As shown in Fig. 1R, coarser finite element/volume method (FEM/FVM) meshes are used to solve the temperature field on the melt pool scale, while cellular automaton (or phase field) approach is performed on finer grids using the temperature field interpolated from the coarser ones. Although the method has done well in the grain structure prediction, it could hardly establish the interactions of solute elements between the solidified microstructural scale and the melt pool scale.

Fig. 1R The schematic diagram of multi-grid method used in the grain structure simulation^{3,4}.

Even some high-fidelity models focusing on the in-situ alloying⁵ and effect of oxygen⁶ account for the solute transport in the melt pool, the solute partition during the solidification process is still not well described. An important aspect, for example, there is a lack of description on how the rejected solute element during solute partition transfers in the melt pool under the effect of melt convection. Eq. R1 used to describe the liquidus temperature of Al-Zr binary alloys could not accurately describe the actual solidification path of superalloys, and there is no source term in the Eq. R2 which is used to describe the solute transport in the melt.

$$T_L = T_L^{Al} \cdot (1 - C_{Zr}) + T_L^{Zr} \cdot C_{Zr} \quad (\text{R1})$$

$$\frac{\partial(\rho C)}{\partial t} + \nabla \cdot (\rho \bar{u} C) = \nabla \cdot (D \nabla C) \quad (\text{R2})$$

Furthermore, in the previous solvers⁷⁻⁹, crystal growth is calculated according to the temperature field, and the grain structure is only a result from thermal history, without considering of solute transport and solute partition. Representative simulated microstructure¹⁰ is shown in Fig. 2R. Without descriptions on the fluid flow, solute distribution, and the consequent constitutional undercooling, the morphology of solid/liquid interface (i.e., planar, cellular, and dendritic) cannot be demonstrated precisely.

In contrast, our efforts on the in-house solvers use one mesh to take into account the microstructural evolution and the multi-physical fields. By including the solute transport and describing the solute partition during the crystal growth coupled with thermal-fluid flow field, it enables microstructure and melt-pool multiple fields interacting each other directly. That is, the melt flow changes the solute profile at the solidification front, making influence on the crystal growth, and the rejected solute elements during crystal growth can also be transferred to the bulk region of the melt pool.

Fig. 2R Representative simulated grain structure¹⁰ using the one-way coupling multi-grid method.

We are sorry that initially we did not clarify the novelty of the developments on the solver clearly. We have added detailed explanation on the novelty of our in-house CFD codes (model framework) to the revised Introduction and Method section.

2) The term "macroscale grain structure" is used primarily to reflect effects without segregation, please utilize something less ambiguous (p. 4 and 23, for example).

Response:

Thank you for the comments and sorry for our improper statements. The term “macroscale grain structure” is really ambiguous, as it cannot reflect the effects without solute segregation directly. We have replaced the term with more descriptions for better illustrations in the revised manuscript. For the 2 places where the term was used, they have been modified as:

“However, most of the simulations focusing on the grain structure are based on thermal history only¹¹⁻¹³, and lack of consideration of the elemental distribution”.

“the factors including the effect of melt convection and solute partition on the solute distribution, and thus on the crystal growth, have always been neglected”

3) Have you quantified effects with and without convective contributions? A comparison or even some remarks would greatly add to the value and demonstrating the significance of this work.

Response:

Thank you for the valuable comment and sorry we did not demonstrate the contributions of melt convection clear enough. As the spreading of melted powder and evolution of melt pool free surface are also included in melt convection, the assumption of no convection condition would be unreasonable in the simulation cases of our manuscript. More importantly, melt convection is one of the essential natures in additive manufacturing. And if it were not taken into account, for example, without the strong mixing effect induced by melt convection, the solute profile in the cellular trunks and intercellular regions of the substrate or the previous track/layer would still directly persist, rather than well mixed to be relatively uniformly distributed. In this case, the solute profile of the melted region would still make great influence on and even directly determine the solute distribution and microstructure in the new melt pool. Thus, the simulation results would be far from the actual process and unable to provide valuable

references, if we performed another case without convection for comparison. So, we just make attempts to clarify the convective contributions by making some remarks.

According to the fundamental solidification¹⁴, an analytical Gulliver-Scheil model is proposed to describe the effect of melt convection on the solute profile parallel to the crystal growth direction. Fig. 3R-a show the schematic diagram of the employed analytical Gulliver-Scheil model. Under the “sufficient diffusion in liquid” condition, the solute in the front of the solid/liquid interface could be well mixed with the melt by the strong thermal-fluid melt flow. In contrast, under the ‘no convection’ condition, there is a solute boundary layer at the solidification front. Melt flow intensity determines the thickness of solute boundary and the solute concentration at the solid/liquid interface. However, under the rapid crystal growth condition, the solute boundary layers can also be suppressed due to the solute trapping effect, as shown in Fig. 3R-b.

The analytical model is then employed to the contribution of the two factors. Assuming the solid/liquid interface could reach liquidus (T_{liq}) all the time, such as from time n to $n+1$, cooling rate (\dot{T}) can be regarded as the increase in crystal growth undercooling degree (ΔT) in a unit time (Δt), as shown in Eq. R3. At the solute boundary layer, the interfacial solute concentration (C_l) equals to the division of the initial concentration (C_0) and the partition coefficient (k). And the interfacial velocity (V) is calculated using Lipton-Glicksman-Kurz (LGK) model, as shown in Eq. R4 (where α_1 , α_2 , and α_3 are fitted growth kinetic constants⁴). Then the partition coefficient can be calculated based on the rapid solidification model established by Aziz et al¹⁵, as shown in Eq. R5, where V_d is critical interfacial velocity and k_e is equilibrium partition coefficient. The non-equilibrium partition coefficient can be used to evaluate microstructure and solute profile. Based on Eqs. R3-R5, iterations were conducted to achieve the solution of the variables.

Fig. 3R-c demonstrates the comparison of the two cases and quantify the contributions of melt convection under the different crystal growth undercooling degrees, which are the deviation between local and liquidus temperatures. Here, the crystal growth undercooling degree on the x -axis is the one in the no convection case. For the Nb element in Inconel 718 (IN718), under

a relative low cooling rate (close to the equilibrium state), the melt convection can provide extra crystal growth undercooling degree as large as about 57 K. During the same thermal history, this promotes the solute trapping effects and can raise the partition coefficient from 0.48 to as high as 0.7.

$$T^{n+1} = T^n - \dot{T}\Delta t \quad T_{liq} = T_{melt} + mC_l \quad \Delta T = T^{n+1} - T_{liq} \quad (R3)$$

$$V(\Delta T) = \alpha_1\Delta T + \alpha_2\Delta T^2 + \alpha_3\Delta T^3 \quad (R4)$$

$$k = \frac{k_e + V/V_d}{1 + V/V_d} \quad C_l = C_0 / k \quad (R5)$$

Fig. 3R Effect of melt convection on the solidification behaviours based on the analytical Gulliver-Scheil model. **a** Schematic diagram of solute boundary layer at the solidification front. **b** The solute profiles under different crystal growth conditions. **c** Convective contribution to the solute trapping effect under different crystal growth conditions.

To further verify the remarks above, a representative numerical simulation of solidification microstructure without convection¹⁶ is cited here to illustrate the contribution of melt convection. In their work, the temperature and flow fields are simulated using a macro-scale model. Also, the characteristic temperature gradient and cooling rate are then applied as

boundary conditions in the microstructural simulation. Fig. 4R shows their simulation results¹⁶ and the experimental data¹⁷ under similar operating conditions. The (quasi) solute trapping region is about half the height (5 μm in difference) of the experimentally measured one. The differences could indicate the convective contributions quantitatively, and confirm the melt flow plays a rather important role in the evolution of microstructure and solute transport. As it is also shown in Fig. 1, the melt convection levels the difference in solute concentration between the cellular trunks and the intercellular regions. Besides, the strong melt flow takes the rejected solute in the front of the solid/liquid interface, accelerating the local solidification rate. We have summarised the remarks above and added them to the revised Supplementary Materials.

Fig. 4R Simulated microstructures without considering convective contributions¹⁶ and the microstructure observed in the reported experiments¹⁷.

4) The non-equilibrium nature is not particularly unexpected here but widely reported in earlier works (line 170 forwards) - in your treatment of, e.g., trapping is there anything you would see differentiating your work from earlier ones?

Response:

Thank you for your comments. Before Line 170, the microstructure evolution and solute transport occur during printing the first layer on the substrate, but additive manufacturing process is related to repeatedly cooling and heating the new layers on the previous layers consecutively with the ultrahigh cooling rate of $10^5 \sim 10^7$ K/s. The phenomenon of re-melting and re-solidification including intrinsic heat treatment (heating layers below due to heat diffusion) can also occur. Thus, it is of great important to understand what happens in the previously printed layer and in the newly printed layer after the subsequent layer printing. It is also crucial to confirm whether the non-equilibrium behaviour could be still reserved after the multi-layers printing. Special attention has been paid to the non-equilibrium nature of additive manufacturing, because it deserves discussing whether we could gain insight into this phenomenon to reduce the occurrence of crack susceptibility. In nickel-based superalloys, we want to demonstrate the difference of non-equilibrium behaviours in representative three superalloys, and explain the difference in crack susceptibility among the three superalloys in terms of solute segregation: CM247LC (higher γ' superalloys with high strength), IN718 (lower γ' superalloys) and ABD850AM & ABD900AM (newly designed superalloys optimised for AM). It would be valuable if we improved the built part without sacrificing any high-temperature mechanical performance.

Sorry we did not show a clear streamline for the manuscript. We have added necessary statements at the beginning of Discussion in the revised manuscript to make the streamline more coherent and more visible.

5) The binary (dilute-like) treatment of the multi-elemental system, have you made any sensitivity analyses or addressed and investigated any possible effects which might not be captured due to this approximation?

Response:

Thank you for the comment. We are sorry that we did not describe our model clearly enough. Dilute binary treatment is used to consider the multi-element alloy system, neglecting the effects of chemical activities, chemical reactions, element interdiffusion, and the change in

solidification path. According to the previous works on the modelling of solidification^{18,19}, we believe these assumptions are generally acceptable. The changes in partition coefficient of each element can also be regarded as the interactions among the solute elements. Besides, the solution process would be rather complicated and unstable, if all these factors were included in this model framework. Under the consideration of the computational capacity, we thought it might be better to make these necessary simplifications. Sorry we did not include these necessary descriptions in the first place. In the section of “Model description on solidification of the multicomponent alloy system” in the revised Supplementary Materials, we have clarified the limitation of this work to prevent any ambiguity. Besides, we should use “thermal-fluid-solutal” as a subtitle, rather than “thermal-solutal-chemical”. The latter one might be confusing and misleading, due to too many aspects are included in this broad concept. By using the “chemical” term, what we wanted to express is the solute transport and partition of the solute elements. We have amended the subtitle in the revised manuscript accordingly.

6) The powder bed looks like it has a rather low fraction of powder, how was this determined and you see this as realistic and not influencing the results?

Response:

Thanks for your comments. The powder distribution is determined from discrete element method, as illustrated in the previous publication²⁰ using the data of powder diameter measured experimentally. The three-dimensional powder distribution is shown in Fig. 5R. Compared with the commonly reported powder distributions, we believe such a fraction is representative to the powder size distribution in real scenario. Due to the packing density is limited on the first layer, the fraction of powder on a certain section looks a bit low, as seen from the black section. From this aspect, although powder bed looks like low in the fraction of powder, we could still consider it as reasonable and representative. We are sorry that we did not provide the necessary information in the first place, and we have added the description to the revised Supplementary Materials to make our treatment more rigorous.

Fig. 5R Three-dimensional powder distribution and the longitudinal section.

7) How was the initial microstructure and simulation initialized?

Response:

Thanks for your comments. We are sorry that we did not demonstrate the way we reached the actual initial condition of microstructure simulation clearly. Initially, we marked the areas of the substrate and the powder, and then set them to be filled with metallic phase with initial composition and liquidus temperature. Then, we simulated the grain nucleation and growth in the substrate and the powder under the condition ($h = 80 \text{ W/m}^2\text{-K}$, freestream temperature of 300 K) commonly used in conventional casting process. After the solidification process, the temperature of the domain was reset to room temperature. Thus, we got the solute distribution and grains in the substrate and powder as the initial conditions for the LPBF process.

We have added a new sub-section “Simulation parameters” in the Method section, and moved the information on how the initial microstructure was initialized from the “Model verification” here, to clarify this point in the revised manuscript.

8) The analysis is 2D essentially? What about possible implication of 3D effects in the results.

Response:

Thank you for the valuable comment. Yes, the analysis is 2D essentially. Due to the strongly coupling nature between thermal-solutal-fluid flow and solidification model, we selectively

choose the 2D plane from three-dimensional (3D) powder distribution model to simulate the detailed solidification with thermal fluid flow and solute transport in three different AM superalloys. Based on our simulation results and the reported experimental characterisations²¹ and 3D simulations^{22,23}, we would like to discuss the implication of 3D effects here. According to the reported 3D flow field analysis on the melt pool of LPBF process, the Marangoni effect drives a strong flow circulation from the laser spot to the tail of the melt pool along the scanning path in a long distance, as shown in Fig. 6R.

Fig. 6R Representative three-dimensional flow field in the melt pool of LPBF process^{21,22}.

The melt flow parallel to the scanning direction would scour the solidification front at the bottom of the melt pool at the tail, transferring the solute elements rejected by the cellular (dendritic) trunks to the forward of the melt pool of higher temperature. This could contribute to the relatively stable composition in the liquid melt during the solidification sequence. Besides, for the melt pool geometry, the length in the scanning direction is much higher. The melt pool would be more stable with less swaying on the perpendicular section in the real 3D case. Consequently, there would be less transitions and unexpected evolutions of the solute distribution and microstructure induced by the perturbation of melt flow.

More importantly, considering the melt flow parallel to the scanning path, laser beam could still make influence on the solidification process even after it has passed away. Though the laser spot has passed away, the local heat in the tail of melt pool could still be compensated by the forward high-temperature melt, which prolongs the solidification process and reduces the

cooling rate. *This effect leads to the much wider gap in approaching the (quasi) solute trapping non-equilibrium state.* Thus, an optimisation of heat source (laser beam) profile could be a possible route to achieve a better thermal condition in the LPBF of superalloys.

We have carefully made discussion on these possible implication of 3D effects in the Discussion section of the revised Supplementary Materials.

9) "The microstructure in the overlapping zone is greatly improved after the re-solidification" (line 161)- what is actually improved and where does this relate to?

Response:

Thanks for your comment and we have improved the statement to make it clear. It is ambiguous to simply describe it as “improved microstructure”. We should have explained in which aspects it can be considered as “improved”. To be more precise, it is the solute segregation pattern that is improved. Severe intercellular solute enrichment is formed after the first-track printing, as the marked in Fig. 2d. That is the aspect we consider it is “improved” after re-solidification.

We have replaced the original statement with “intercellular solute enrichment”, and added essential explanation for the “improved” in the revised manuscript.

10) Solute trapping driven suppression of cracks, can you make any mechanical arguments based on, e.g., the specific behavior of the studied alloys as to why this could be a benefit?

Response:

Thanks for your valuable comments. Firstly, we did not make enough introduction on how the cracks make negative influence on the mechanical performance of superalloy components. Cracks can greatly accelerate the failure/rupture of the superalloy components under the high-temperature and high-pressure working condition. To improve the working performance, research focusing on the inhibition of cracks are conducted to prolong the creep rupture life and fatigue life. However, in the additive manufacturing of superalloys, cracks are always formed just in as fabricated or as heat treated condition, before the component being applied, greatly limiting the application of this future technology of nickel-based superalloys. In the revised

Introduction, we have added these statements on the detrimental effect of crack in the as-built parts to better illustrate the significance of this work.

Secondly, in terms of the mechanical properties, the solute trapping also benefit to the built parts in some other aspects. Grain boundary is weak under the high-temperature working condition, and cracking due to creep degradation tend to occur along the grain (or sub-grain) boundaries. For the nickel-based superalloys without grain boundary strengthening elements, the lower level of segregation caused by solute trapping effect reduces the grain boundary brittleness to suffer the high thermal stress induced by the thermal cycles²⁴. In the subsequent heat treatment process for the as-additively manufactured parts, the reduction of elemental segregation can diminish the size difference of precipitated γ' phase between the cellular trunk and intercellular region, and thus to lessen the stress concentration during the precipitation process. Besides, the uniformly distributed small γ' phase can also further prolong the rupture life of the superalloy components. Due to the rapid solidification in AM (typically 10^5 - 10^7 K/s), the non-equilibrium microstructure such as cellular structure can be induced. The relation of solute distribution and the cellular structure should be studied to understand the heterogeneity in as-printed parts, and subsequently we can tailor the microstructure using the optimised process conditions and strategic heat treatment protocol.

We have added these arguments on the way solute trapping could benefit to the mechanical properties of superalloy components to the revised Discussion.

11) The level of validation in the work, is there validation beyond the given literature sources? Also, the validation seems to point primarily towards bead sizes and shapes, while the results relate to segregation and trapping. Could you provide information on how the method compares, e.g., in terms of the predicted elementary distribution or microstructural features directly?

Response:

Thanks for your thought-provoking comments. To further validate the model framework in more details, we have performed experiments to characterise the microstructure and to measure the elemental distribution profile in the cellular trunks and intercellular regions. An IN718

sample of 10 mm × 5 mm × 5 mm (length-width-height) was LPBF-ed under the same operating parameters, as per Ref. 70. The sample was then sectioned longitudinally perpendicular to the scanning direction. Fig. 7R shows the representative cellular structure observed using the backscattered electron (BSE) on a scanning electron microscope (TESCAN MIRA 3). The observed characteristics of the microstructure is compared well with the simulation results.

Fig. 7R Cellular structure observed on the longitudinal section of the as-built IN718 sample.

Then, a wavelength dispersive spectrometer (WDS) equipped on an electron probe microanalyser (EPMA-8050G, Shimadzu) was employed to quantify the solute profile across the cellular trunks and intercellular areas with the spatial resolution of 100 nm. Fig. 8R shows the comparison of the mass fraction profile of Nb, the element of the most focus with strong segregation tendency in the superalloy IN718. It could be concluded that for similar primary dendrite (cellular) arm spacings (PDAS), the simulation results and experimental data reach a reasonable agreement.

To further verify the simulated solute trapping effect qualitatively, the elemental profiles of the cellular structures of different PDAS were characterised using scanning transmission electron microscopy and energy dispersive X-ray spectroscopy (STEM-EDS, Talos F200X). Fig. 9R demonstrates the Nb concentration distributions across the cellular and intercellular areas. The segregation range, that is the oscillation of solute profile, decreases with the refinement of the cells. Thus, these could further verify the simulation results and discussion in terms of the solute trapping effect.

Fig. 8R Comparison of experimental data (EPMA-WDS) and simulation results on the Nb distribution.

Fig. 9R Effect of solute trapping effect on the segregation profile and primary dendritic (cellular) arm spacing.

We have summarised the verification of microstructure and solute profile shown above in the revised Fig. 6. The results and the corresponding descriptions have also been added to the

Method section (Model verification: precision and high-fidelity). We hope this could help verify our model framework directly. In order to keep the streamline of the manuscript, we have demonstrated the details of the experiments in the revised Supplementary Materials.

Reviewer #2:

The concept of this work is novel and interesting. The idea of coupling melt pool modelling, fluid flow, and solute trapping during additive manufacturing is unique. The work has the potential to enable guidelines for determining printability of materials in a predictive manner rather than the empirical approaches in use currently. However, the paper, in its current forms has several shortcomings that should be addressed before being considered for publication:

Response:

The authors appreciate the reviewer for the kind compliments and valuable comments. Please find the point-by-point responses we have made carefully to address all the comments.

Comments:

1) The abstract should be modified to highlight the uniqueness of the work as well as key results. In the current form, the abstract appears to be a summary of "what was done" rather than "key findings of the paper". This change will be important to intrigue and excite a broader audience to read the paper.

Response:

Thanks for your valuable comments. We have made efforts to improve our abstract to show more highlights (including the uniqueness of our methodology, key findings) of this manuscript. We hope these could help intrigue and excite more readers.

2) The manuscript, in general, is very difficult to read and appears to jump from one concept to another. Certain sentences appear incomplete or unclear. There is use of jargon or overly complex sentence construction. As an example, the authors use metal pool instead of the more common term "melt pool". I would suggest the authors to make the manuscript more

streamlined and succinct in nature. It will enable the communication of a clear-concise message. In the current form the novel needs to be searched for vs. streamlining it will make it clearer. The attached document has more specific comments and some suggested corrections. As an example, the authors use thermal-solutal-chemical-multiple physics as the first section of results. This is quite confusing and does not quite inform the reader, even those familiar with metal AM and modelling, as to what this section is about.

Response:

Appreciate your valuable comments on the writing details. We are very appreciated such thought-provoking comments and suggestions in the attached document. They do help us improve the manuscript a lot, and we are very grateful for your contributions sincerely. As it is commented, manuscript is hard to read, with the concepts always jumping from one to another, and some complex and confusing sentences. Sorry for these shortcomings. We have made efforts to improve our writing and streamline the overall manuscript: replacing terms with more common ones (such as “melt pool”), adding necessary transition statement, deleting the controversial or improper statements, and re-arranging abrupt paragraphs, using more easy-to-understand descriptions (such as “thermal-fluid-solutal transport”, the sub-title for the first section), and providing explanations to the concepts. We have responded all these detailed valuable comments and suggestions point-by-point in the revised manuscript, and made the corresponding modifications and marked them in yellow in the revised manuscript accordingly.

3) In the introduction, the authors claim that the use of convection is still always neglected. This is incorrect since T. Debroy and Wei Zhang have focused on these aspects in their works to name a few. The authors also say that they have developed a two-way fully-coupled mathematical model. Can they clarify, what does two-way mean here?

Response:

Thank you for the comment and sorry for our improper statement. It is incorrect to simply say “the role of convection is still always neglected”, because the effect of melt flow on some aspects has been considered in their works. For example, as the reviewer commented, Debroy

and Zhang² reviewed the recent progress on the modelling of additive manufacturing, and the effect of melt convection on the temperature field and thus grain structure has already been discussed. We have modified the original statement to make it more specific and rigorous.

However, some physical effects are still neglected in the previous investigations including:

- a. The effect of melt convection on the solute distribution;
- b. The effect of solute distribution on the crystal growth;
- c. The effect of solute partition on the solute distribution in the melt pool;

Two-way coupling approach is a concept to improve the “one-way” coupling method proposed by Yan et al.^{7,25}, where the temperature and flow fields are solved and then exported as an initial condition to simulate the solute transport and dendritic growth on the microstructural scale. For the claimed “two-way coupling”, it means the transport phenomenon (including temperature and solute fields) can determine the solid/liquid interfacial evolution, and the microstructure is also able to change the flow pattern in the melt pool in reverse. Besides, the melt convection also changes the local composition of the melt and thus makes influence on the grain structure, and the solute partition on the microstructural scale can in reverse change the solute transport in the melt pool. These interactions of two directions can hardly be described in the “one-way” coupling method.

Sorry it is too abrupt to directly jump to this concept, and we did not explain in what way the model is named as “two-way” in the Introduction. We have amended it by replacing it with “strongly coupling” and illustrating the “two-way” in the Method section, to clarify the uniqueness of our model framework in the revised manuscript.

4) In the results section the authors start with "with the well validated model", yet there is no direct reference to either the supplementary section or a figure therein that can guide a reader to figure out how was the model validated? Where was the validation data obtained from? Granted some of these aspects are discussed later, but this style of writing breaks the flow and raises more questions than it addresses answers.

Response:

Thank you for the comments. We agreed that to start with “with the well validated model” is quite confusing, because there is no model validation before the statement. We have rewritten the sentence by adding a direct reference for readers to find out the details on the framework in the Method section. Similarly, as it is commented here and in the attached document, we did improve in streamlining the manuscript. We have made the corresponding corrections to keep the writing flow of the manuscript and avoid raising extra questions.

5) In figure 1, the color map is difficult to read. Further, as the melt pool movement is shown it would be beneficial to also highlight the velocity vectors since it has a significant role in the melt pool dynamics.

Response:

Thanks for your valuable comments. For a better visualisation, we improve the colour legend and provide a high-resolution version of the figure with a much larger size with the revised manuscript submission to show more details of Figure 1. We have made attempts to show the velocity vectors in the sub-figures of solute distribution, but we found they made the solute distribution hard-to-read. Thus, we used scaled arrow heads to demonstrate the melt pool dynamics instead. For a better view, we have also zoomed in on the melt pools of Fig. 1 and added the close-up views to the revised Supplementary Materials (Supplementary Figure 1). Please find the revised Figure 1 in the revised manuscript and the high-resolution version at the end of the merged PDF file.

6) Negative segregation is used as a key concept. However, it is not clear as to what it means or how is it defined? Also, the use of slight negative segregation leaves ambiguity about how much is slight and how much is more. It is recommended that such qualitative assessment be avoided for clarity.

Response:

Thanks for your comments and sorry we did not define the concept clearly enough. Negative segregation is defined as the local concentration of the solute element which is less than the

initial one (for the elements whose partition coefficient is less than 1, and vice versa). In other words, $SI < 0.0$. The use of the adjective “slight” is quite ambiguous, because it could not provide exact quantitative information as we expected. We have rewritten the corresponding statements and tried to avoid such qualitative assessment as much as we could.

7) In Figure 3, the segregation index data is not normalized and the minimum and maximum values in the color band are not the same so it is difficult to quantify the differences visually. Also, it is recommended to add the reference for the experimental data in the figure caption. While the authors claim that the model agrees with the data, has the model been normalized with respect to the process variables (power, velocity, hatch) used for the experimental data? If not, what is the impact of these variables? Further, it is not clear how the experimental data is corroborating the findings in the absence of discussion on the role of process variables.

Response:

Thanks for your comments. Actually, we have considered to use the same colour band, because it would be much easier to compare the differences among alloy systems. However, the segregation tendency of certain solute elements is rather different in quantity, as indicated by the range of the colour bands (± 0.20 , ± 0.34 , and ± 0.84). We could hardly get the information on the solute distribution from the other two contours if the same colour map were used. Under the consideration above, we chose to use the colour bands of different ranges to better show the characteristics of the solute distributions, and the difference in quantity can still be indicated by the range.

For the cited experimental results in sub-figure c, f, and i, except for the figure captions of the sub-figures, we have also added the references to the main figure caption in the revised manuscript. And we have enlarged the size of Fig. 3 and provided a high-resolution version of Fig. 3 submitted with the revised manuscript. In addition, we have also drawn the close-up views of the experimental results in the revised Supplementary Materials (see Supplementary Figure 2). We hope these could make this figure more easy-to-read.

For the process variables, they do play crucial roles as you commented, but we discuss the results on effect of superalloy compositions into the printability. The range of available process variables is generally used for estimating the printability of alloys too. The critical process window for the “hard-to-print” superalloys is so narrow that the possibility of searching it by parameter study seems rather low. In other words, the crack susceptibility is always rather high under common processing parameters. In contrast, the easy-to-print or printable alloys shows much wider process windows, and the similar solute profile and microstructure could be found under a common range of process parameters. Taking IN718 superalloy for example, the features of the solute trapping regions could be found under a wide process window from 0.1~0.3 J/mm²⁶. The process variables in the cited three experimental results are just marginally different, as shown by the details in the Supplementary Materials (section “Process conditions in the cited experiments”). Thus, we believe the comparisons here could show a qualitative verification to capture the main characteristics of microstructure, morphology of solute trapping regions, and solute profile near the cracks, respectively. However, the parameters in the simulation cases are not completely the same as those in the experiments, and it is improper to claim “in good agreement with the data”. Sorry for our careless statements, we have changed them using strict ones to demonstrate a verification of the phenomenon.

8) At the end of the results section, the authors claim that the solute segregation is occurring near the porosity. While one can reach that conclusion from Figure 3i, that is a special case (also, the figure is very hard to read). Most importantly, the authors have not shown or discussed how they have accounted for that in their model? The images from the model seem to show a melt pool with segregation but none of the images have a representation of porosity, lack of fusion or gas porosity. Therefore, at best it is an assertion.

Response:

Thanks for your comments and sorry for the disadvantages. What we actually could conclude is that the solute segregation is occurring in the last-to-solidify region. However, our statement is too generic and misleading, and the readers would consider the solute segregation occurs near

all the porosities in the printed layer. Not all the porosities are the last to solidify, so solute segregation is not always occurring near the porosity. Thus, our discussion on this aspect is incorrect, and we have deleted the relevant statements.

For the model description on the porosity, we did not consider the formation of vapour and the subsequent formed gas bubbles in our framework. We use the volume of fluid method to track the interface between liquid melt and protective gas (argon). During the melting, solidification, and the oscillation of the gas-metal interface, a certain volume of gas is possible to be trapped in the liquid melt and unable to escape until the end of solidification. That is the porosity we accounted (or it is better to call it “process-induced porosity/void” or “fluid flow induced porosity/void”). We have clarified the concept of the mentioned porosity and added the description on how the porosities (voids) are considered to the Method section.

9) In the methods section, while describing fluid flow, the authors call it two fluid phases of metal and gas. It should be simplified to molten metal and gas.

Response:

Thanks for your comments and sorry we did not illustrate our model description clearly. As you commented, it is confusing to call it “two *fluid* phases of metal and gas”. Due to both molten metal and solid metal are treated as a single phase in the multiphase framework, we are still afraid that it might be confusing whether we treat the solidified metal as another phase if we simplify it to be molten metal and gas. So, we have changed it to be “metallic phase and gaseous phase” and added more descriptions on the molten metal and solidified metal to avoid confusion.

References

1. DebRoy, T., Mukherjee, T., Wei, H. L., Elmer, J. W. & Milewski, J. O. Metallurgy, mechanistic models and machine learning in metal printing. *Nat. Rev. Mater.* **6**, 48–68 (2021).
2. Wei, H. L. *et al.* Mechanistic models for additive manufacturing of metallic components. *Prog. Mater. Sci.* **116**, 100703 (2021).

3. Gandin, C. A. & Rappaz, M. A coupled finite element-cellular automaton model for the prediction of dendritic grain structures in solidification processes. *Acta Metall. Mater.* **42**, 2233–2246 (1994).
4. Lian, Y. *et al.* A cellular automaton finite volume method for microstructure evolution during additive manufacturing. *Mater. Des.* **169**, 107672 (2019).
5. Samaei, A., Sang, Z., Glerum, J. A., Mogonye, J.-E. & Wagner, G. J. Multiphysics modeling of mixing and material transport in additive manufacturing with multicomponent powder beds. *Addit. Manuf.* **67**, 103481 (2023).
6. Chia, H. Y., Wang, L. & Yan, W. Influence of oxygen content on melt pool dynamics in metal additive manufacturing: High-fidelity modeling with experimental validation. *Acta Mater.* **249**, 118824 (2023).
7. Yang, M., Wang, L. & Yan, W. Phase-field modeling of grain evolutions in additive manufacturing from nucleation, growth, to coarsening. *npj Comput. Mater.* **7**, 56 (2021).
8. Yu, Y., Li, Y., Lin, F. & Yan, W. A multi-grid Cellular Automaton model for simulating dendrite growth and its application in additive manufacturing. *Addit. Manuf.* **47**, 102284 (2021).
9. Xiong, F., Gan, Z., Chen, J. & Lian, Y. Evaluate the effect of melt pool convection on grain structure of IN625 in laser melting process using experimentally validated process-structure modeling. *J. Mater. Process. Technol.* **303**, 117538 (2022).
10. Shi, R. *et al.* Microstructural control in metal laser powder bed fusion additive manufacturing using laser beam shaping strategy. *Acta Mater.* **184**, 284–305 (2020).
11. Sunny, S., Yu, H., Mathews, R., Malik, A. & Li, W. Improved grain structure prediction in metal additive manufacturing using a Dynamic Kinetic Monte Carlo framework. *Addit. Manuf.* **37**, 101649 (2021).
12. Pinomaa, T., Lindroos, M., Walbrühl, M., Provatas, N. & Laukkanen, A. The significance of spatial length scales and solute segregation in strengthening rapid solidification microstructures of 316L stainless steel. *Acta Mater.* **184**, 1–16 (2020).

13. Han, Y., Griffiths, R. J., Yu, H. Z. & Zhu, Y. Quantitative microstructure analysis for solid-state metal additive manufacturing via deep learning. *J. Mater. Res.* **35**, 1936–1948 (2020).
14. W. Kurz; Fisher, D. J. *Fundamental of solidification*. (1984).
15. Aziz, M. J. Model for solute redistribution during rapid solidification. *J. Appl. Phys.* **53**, 1158–1168 (1982).
16. Wang, Y., Shi, J. & Liu, Y. Competitive grain growth and dendrite morphology evolution in selective laser melting of Inconel 718 superalloy. *J. Cryst. Growth* **521**, 15–29 (2019).
17. Yi, J. *et al.* Microstructure and mechanical behavior of bright crescent areas in Inconel 718 sample fabricated by selective laser melting. *Mater. Des.* **197**, 109259 (2021).
18. Kurz, W., Rappaz, M. & Trivedi, R. Progress in modelling solidification microstructures in metals and alloys. Part II: dendrites from 2001 to 2018. *Int. Mater. Rev.* 1–47 (2020) doi:10.1080/09506608.2020.1757894.
19. Dantzig, J. A. & Rappaz, M. *Solidification*. (Taylor & Francis Group, 2009).
20. Panwisawas, C. *et al.* Mesoscale modelling of selective laser melting: Thermal fluid dynamics and microstructural evolution. *Comput. Mater. Sci.* **126**, 479–490 (2017).
21. Hooper, P. A. Melt pool temperature and cooling rates in laser powder bed fusion. *Addit. Manuf.* **22**, 548–559 (2018).
22. Lee, Y. S. & Zhang, W. Modeling of heat transfer, fluid flow and solidification microstructure of nickel-base superalloy fabricated by laser powder bed fusion. *Addit. Manuf.* **12**, 178–188 (2016).
23. Tang, C., Tan, J. L. & Wong, C. H. A numerical investigation on the physical mechanisms of single track defects in selective laser melting. *Int. J. Heat Mass Transf.* **126**, 957–968 (2018).
24. Reed, C. R. *The Superalloys: Fundamentals and Applications*. (Cambridge University Press, 2006).

25. Yu, Y. *et al.* Impact of fluid flow on the dendrite growth and the formation of new grains in additive manufacturing. *Addit. Manuf.* **55**, 102832 (2022).
26. Yi, J. H. *et al.* Effect of laser energy density on the microstructure, mechanical properties, and deformation of Inconel 718 samples fabricated by selective laser melting. *J. Alloys Compd.* **786**, 481–488 (2019).

REVIEWER COMMENTS

Reviewer #1 (Remarks to the Author):

Feedback by comment #:

- 1) Although some of the discussed features are contained in earlier works, the authors demonstrate good points on the novelty which are valid.
- 2) The comment has been addressed.
- 3) Thank you for the detailed answer, believe this also addresses nicely the importance of present work and respective implications.
- 4) Ok.
- 5) This is certainly an aspect that is addressed similarly by earlier works.
- 6) & 7) The provided additional details clarify and address the comments and the execution is valid.
- 8) This makes sense, hopefully 3D will be more feasible in the future as is expected, the discussion on effects and its incorporation is satisfactory.
- 9) & 10) The explanation makes sense, still of course we can envision cases where trapping and segregation can yield also non-desirable outcomes which has underlying the comment. The added arguments make the comments specific and see them then as acceptable.
- 11) The added results support the provided argumentation.

Overall, the authors have provided additional results, explanations and demonstrated that the problem definition and selection of respective approaches are suitable.

Reviewer #2 (Remarks to the Author):

I believe the authors have addressed all the comments in a satisfactory manner and the paper can be accepted.